# MULTI-SCALE PROTEIN LANGUAGE MODEL FOR UNIFIED MOLECULAR MODELING

## ABSTRACT

Protein language models have shown great potential in protein engineering. However, the current protein language models mainly work in the residue scale, which cannot offer information in the atom scale. The strong power of protein language models could not be fully exploited to benefit the applications that cross protein and small molecules. In this paper, we propose *ms*-ESM (multi-scale ESM) to realize the multi-scale unified molecular modeling by pre-training on multi-scale code-switch protein sequence and describing relationships among residues and atoms with a multi-scale position encoding. Experimental results show that *ms*-ESM outperforms previous methods in protein-molecule tasks and is on par with the state-of-the-art in protein-only and molecule-only tasks.

## 1 INTRODUCTION

Protein language models (PLMs) have shown great potential in protein engineering, which captures biochemical and co-evolutionary knowledge in the pre-training of large-scale protein sequences, and gives strong results in protein structure prediction (Wu et al., 2022; Fang et al., 2022b), protein fitness prediction (Mardikoraem & Woldring, 2023; Notin et al., 2022), protein design (Zheng et al., 2023; Ferruz et al., 2022), etc. For example, several important models have been built upon ESM (Rives et al., 2021; Lin et al., 2022b)–a widely used PLM–including ESM-fold (Lin et al., 2023) for accurate protein structure prediction and lm-design (Verkuil et al., 2022; Hie et al., 2022) for designing proteins with specific functions.

Current PLMs mainly work in the *protein residue* (amino acid) *scale*, which cannot offer information in the *atom scale*. In such a case, the strong power of PLMs could not be fully exploited to benefit the applications across macro-molecule (protein) and small molecules,[1] and external small molecular models have to be included to deal with these applications. However, proteins are also composed of atoms, and modeling protein only in the residue scale could be of low resolution. Intuitively, extending PLMs to work in both residue and atom scales would make it applicable to a larger range of applications.

However, developing multi-scale PLMs is non-trivial. First, the *unified molecular modeling* that works in both residue and atom scales is infeasible, due to the incompatible vocabularies used in the two different scales. A direct way to inject atomic information into residue scale PLMs is to represent and pre-train the proteins in the atom scale, in addition to the original residue scale pre-training. Nevertheless, a protein can consist of thousands of residues and thus contains hundreds of thousands of atoms, which is quite inefficient for modeling. Second, designing appropriate position encoding to accurately describe the relationships among residue and atoms in the same protein is also challenging, which is quite complex and involves relationships varying from residues to residues, residues to atoms, and atoms to atoms.

To address the above challenges, in this paper, we propose *ms*-ESM (multi-scale ESM), which realizes the multi-scale unified molecular modeling by a) pre-training on multi-scale *code-switch protein*

---

[1]These applications widely exist in chemistry and biology and are always quite crucial for specific scientific discoveries. For example, drug discovery aims to find small molecules that can bind to protein pockets (Anderson, 2003; Batool et al., 2019) and enzyme engineering searches enzymes (a special protein) that can catalyze molecular reactions efficiently (Mazurenko et al., 2019; Kroll et al., 2023a).

*sequence* and b) describing relationships among residues and atoms with a *multi-scale position encoding*.

First, inspired by the idea of multi-lingual code-switching in machine translation (Yang et al., 2020; Li et al., 2022a)[2], *ms*-ESM proposes to learn multi-scale knowledge by pre-training on the *multi-scale code-switch protein sequences*, which is obtained by randomly unzipping protein residues into their corresponding atoms. In such a case, *ms*-ESM can not only capture the multi-scale aligned knowledge but also efficiently deal with inputs in both residue and atom scale in the meantime.

Second, *ms*-ESM employs a *multi-scale position encoding* for comprehensively distinguishing residues and atoms in the code-switch protein sequence. In the residue scale, we extend the original position encoding used in ESM to consist with the current best practice in pure residue situations and avoid ill-defined position information among atoms. In the atom scale, to distinguish the relation among unzipped atoms, we use the spatial distance matrix directly encoding their 3D positions. With the above approach, we can appropriately describe all the relationships of all objects in the code-switch sequence.

We use three types of downstream tasks (protein-molecule tasks, protein-only tasks, and molecule-only tasks) to demonstrate the versatileness and effectiveness of our proposed *ms*-ESM. In protein-molecule tasks, *ms*-ESM outperforms previous methods that model proteins and molecules separately instead of unified modeling as *ms*-ESM. In protein-only and molecule-only tasks, *ms*-ESM is on par with the state-of-the-art. Experiment results show that we successfully model proteins and molecules in a unified style without suffering from severe information interference.

## 2 PROPOSED METHOD: *ms*-ESM

In this section, we describe our multi-scale pre-training model, i.e., *ms*-ESM, in detail. Intuitively, inspired by the idea of multi-lingual code-switching method, *ms*-ESM first creates multi-scale code-switch protein sequences by unzipping partial residues. Through training on such sequences with correctly designed multi-scale position encoding, *ms*-ESM can work well in both residue and atom scale. When dealing with protein-molecule tasks, *ms*-ESM does not need any extra models and can exert the maximum potential of pre-training.

Specifically, in Section 2.1, we first introduce the overall objective of training *ms*-ESM. Then, in Section 2.2, we dive into the details about how we construct a code-switch protein sequence and implement the multi-scale pre-training. To describe the complicated position relationship in the code-switch sequence, we design a multi-scale position encoding in Section 2.3. In Section 2.4, we provide more details about *ms*-ESM, including an elaboration of its parameterization.

### 2.1 OVERVIEW

We start with an overview of our multi-scale pre-training model, i.e., *ms*-ESM (see Figure 1). Briefly, the total objective of our pre-training can be written as the following loss function:

$$\mathcal{L}_\theta = \sum_{X_i \in D} \mathcal{L}_{\text{MLM}}(\bar{X}_i, E_i; \theta) + \mathcal{L}_{\text{PDR}}(\bar{X}_i, E_i; \theta)$$

$$= \sum_{X_i \in D} \mathcal{L}_{\text{MLM}}(\text{UNZIP}(X_i), \text{MSPE}(X_i); \theta) + \mathcal{L}_{\text{PDR}}(\text{UNZIP}(X_i), \text{MSPE}(X_i); \theta)$$

For each data $X_i$ in dataset $D$, we first create its code-switch sequence $\bar{X}_i$ by unzipping partial residues. Based on the code-switch sequence, we use Masked Language Modeling (MLM) and Pair-wise Distance Recovery (PDR) as the pre-training tasks. We discuss the details of $\bar{X}_i$, $\mathcal{L}_{\text{MLM}}$, and $\mathcal{L}_{\text{PDR}}$ in Section 2.2. As residues and atoms coexist in the sequence, we further design a Multi-Scale Position Encoding (MSPE) $E_i$ to describe the complicated position relationship in $\bar{X}_i$ (see Section 2.3). We show more details of *ms*-ESM, including the parameterization of $\theta$ in Section 2.4. Notably, as we also use the molecule data in pre-training, *ms*-ESM can take proteins or molecules as input separately.

---

[2]construct sentences that alternate between two or more languages

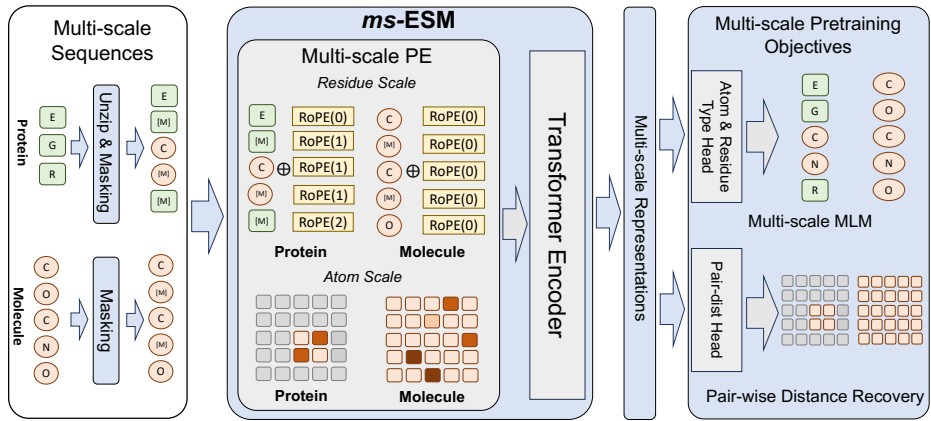

Figure 1: Overview of our multi-scale pre-training process.

## 2.2 MULTI-SCALE PRE-TRAINING

In this section, we elaborate how we create a code-switch sequence $\bar{X}$ and adopt the pre-training tasks, i.e., MLM and PDR on it (see Figure 2).

**Code-Switch Protein Sequence** Specifically, in residue scale, a protein $X$ can be seen as a sequence of $L$ residues, i.e., $X = (r_1, \cdots, r_i, \cdots, r_L)$. Each residue $r_i$ further consists of a specific set of $N$ atoms $A_i = \{a_i^1, \cdots, a_i^N\}$. To create a code-switch protein sequence $\bar{X}$, we first choose a group of residues and insert their corresponding atoms into $X$. Especially, when inserting the atoms, we first assign an order to them. For example, after inserting the atom set $A_i$ into $X$, we get a code-switch sequence

$$
\begin{aligned}
\bar{X} &= (r_1, \cdots, r_i, \text{ORDER}(A_i), \cdots, r_L) \\
&= (r_1, \cdots, r_i, a_i^1, \cdots, a_i^N, \cdots, r_L) \\
&= (h_1, \cdots, h_i, h_{i+1}, \cdots, h_{i+N}, \cdots, h_{L+N})
\end{aligned}
$$

where ORDER is the order that is assigned to atom set (see Appendix A). $h_i$ represents a single residue or atom in $\bar{X}$. We also denote all the atoms in $\bar{X}$ as $\bar{A}$ and all the residues as $\bar{R}$.

Notably, when we insert the atom set $A_i$ of residue $r_i$, we still retain $r_i$. This allows the model can either attend to the corresponding residue scale information or to the surrounding atom scale information when predicting masked atoms, which encourages the model to align residue scale and atom scale representations, just like cross-lingual pre-training (Conneau & Lample, 2019). We show an illustration of the code-switch sequence in Figure 2.

**Masked Language Modeling** After obtaining the code-switch sequence $\bar{X}$, we can do the MLM on it. Different from the MLM used in ESM, we ask models to predict not only masked residues but also masked atoms. Specifically, we first randomly mask part of atoms or residues in $\bar{X}$, and then ask the model to predict the original atoms or residues based on the context.

$$
\mathcal{L}_{\theta\text{MLM}} = - \sum_{h \in \text{MASK}(\bar{X})} \log p_\theta(h | \bar{X} \backslash \text{MASK}(\bar{X}))
$$

where MASK is the set of masked atoms and residues, and $h$ is a single masked atom or residue. Figure 2a is the framework of MLM task.

**Pair-wise Distance Recovery** We also use the PDR as another pre-training task. Briefly, we use corrupted atoms as model input and ask model to recover the correct Euclidean distances between these atoms. We corrupt the atoms by adding noises to their coordinates. Specifically, we use a random position which is around (Euclidean distances $< \epsilon$, Appendix A) the ground-truth coordinate to replace the ground-truth. Models need to recover the real distances based on the corrupted coordinates.

$$
\mathcal{L}_{\theta\text{PDR}} = \sum_{\substack{h_i, h_j \in \bar{A}, i \neq j \\ c_i = \text{COORD}(h_i) \\ c_j = \text{COORD}(h_j)}} \|\text{DIS}_\theta(c_i + \sigma_i, c_j + \sigma_j) - \text{DIS}(c_i, c_j)\|_2
$$

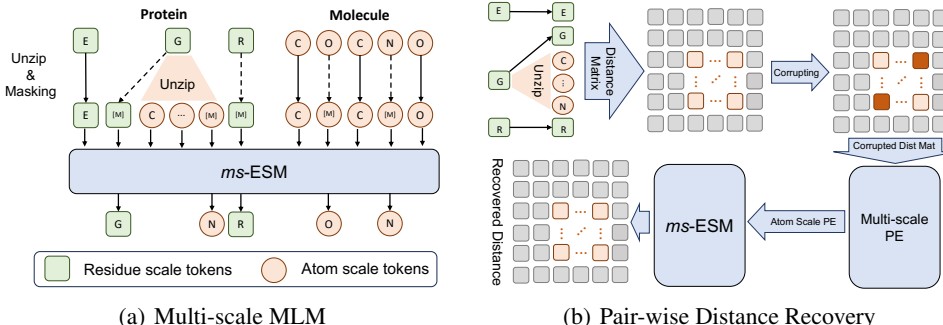

(a) Multi-scale MLM        (b) Pair-wise Distance Recovery

Figure 2: Framework of multi-scale pre-training, which is consisting of multi-scale masked language modeling and pair-wise distance recovery.

where $\text{DIS}_\theta$ is the recovered distance and $\text{DIS}$ is the ground truth. $\text{COORD}$ extracts coordinates from atoms. $\sigma_i, \sigma_j$ are the corresponding noises added to atom coordinates $c_i, c_j$. In more detail, these noises will affect the atom scale position encoding in Section 2.3. Figure 2b shows the framework of PDR task.

Notably, when training $ms$-ESM, we can mix up a protein dataset $D_p$ and a molecule dataset $D_m$ as the final dataset, i.e., $D = D_p \cup D_m$. For a data $X$ from $D_m$, its corresponding $\bar{X}$ is the ordered set of all its atoms and its $\bar{A} = \bar{X}, \bar{R} = \emptyset$.

## 2.3    MULTI-SCALE POSITION ENCODING

Encoding the position relationship in the code-switch sequence is challenging. As residues and atoms coexist in the code-switch sequence, a well-functioning position encoding needs to describe the position relationship from residues to residues, residues to atoms, and atoms to atoms (from the same residue or not). This situation is more complicated than pure residue ones. Because previous encoding in PLM is only designed for pure residue situations, they can not describe the relationship from residues to atoms, and atoms to atoms.

In this section, we design a Multi-Scale Position Encoding $E$ to encode the position relationship in a code-switch sequence. Specifically, $E$ contains a residue scale position encoding $E^R$ and an atom scale position encoding $E^A$, i.e., $E = (E^R, E^A)$. For $E^R$, we carefully extend an existing encoding method letting it can encode the relation from residues to atoms, while keeping consistent with the original encoding when dealing with pure residue situations. For $E^A$, to capture the relationship among atoms, we directly encode their 3D position with the spatial distance matrix. The multi-scale encoding style makes sure that no ill-defined position relationship influences the pre-training letting $ms$-ESM work well in both scales. Figure 3 is the framework of our multi-scale position encoding. We elaborate each of them in the following paragraphs.

**Residue Scale Position Encoding** We design the residue scale position encoding $E^R$ following two principles: a) For encoding the relationship between two residues, $E^R$ should be consistent with the mainstream encoding method. b) For atoms from the same unzipped residue, $E^R$ should not provide any ill-defined position information. As previous PLMs show the success of the mainstream encoding method in dealing with the pure residue situation, it is wise for $E^R$ to keep consistent with it. Moreover, when dealing with two atoms from the same residue, as we can not define the residue scale position relationship inside the residue, $E^R$ needs to avoid the effect of such ill-defined information.

In particular, we use Rotary Position Embedding (RoPE) (Su et al., 2021), the original position encoding in ESM-2, to describe the position relationship among the residues in a code-switch sequence. When we need to assign a position encoding to the atom in the code-switch sequence, we reuse the position encoding of the residue that the atom belongs to. If we can not find which residue that the atom comes from, we assign a fixed position encoding (RoPE(0) in our paper) to it. Formally, for a code-switch sequence $\bar{X}$, its residue scale position encoding $E^R = (e_1^R, \cdots, e_{L+N}^R)$

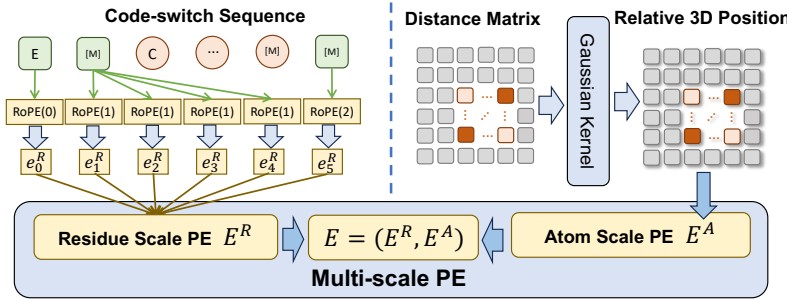

Figure 3: Framework of multi-scale position encoding.

can be obtained according to the following formulation:

$$
e_i^R = \begin{cases} \text{RoPE}(j) & h_i \in \bar{R}, h_i = r_j \\ \text{RoPE}(k) & h_i \in \bar{A}, \exists k, h_i \in A_k \\ \text{RoPE}(0) & \text{otherwise} \end{cases}
$$

By adopting such encoding strategy, $E^R$ satisfies the two principles aforementioned. Specifically, for pure residue situations, $E^R$ is exactly RoPE. When dealing with atoms from the same residue, the relative nature of RoPE makes sure no ill-defined information will affect the pre-training model. We refer readers to Su et al. (2021) for more details of RoPE's properties.

**Atom Scale Position Encoding** Because $E^R$ will not provide the position encoding for atoms from the same residue, we need an atom scale position encoding $E^A$ to describe the relationship from atoms to atoms. As suggested by Zhou et al. (2023), we use Euclidean distance matrix and Gaussian kernel to encode the 3D position of atoms. For $h_i, h_j \in \bar{X}$, their atom scale position encoding $e_{ij}^A$ can be calculate as:

$$
e_{ij}^A = \begin{cases} 0 & h_i \in \bar{R} \text{ or } h_j \in \bar{R} \\ \text{GAUSSIAN}(\text{DIS}(c_i, c_j)) & \text{otherwise}, c_i = \text{COORD}(h_i), c_j = \text{COORD}(h_j) \end{cases}
$$

We refer readers to Zhou et al. (2023) for more details of this 3D position encoding.

### 2.4 OTHER DETAILS OF *ms*-ESM

We parameterize the $\theta$ with a slight modification of the original Transformer (Vaswani et al., 2017). To be specific, we first use our residue scale position encoding $E^R$ to replace the sinusoidal encoding in the Transformer. For the atom scale position encoding $E^A$, we treat it as the bias term of self-attention layers. The self-attention in *ms*-ESM can be calculated like:

$$
\text{ATTENTION}(Q, K, V, E^A) = \text{SOFTMAX}(\frac{QK^T}{\sqrt{d_k}} + E^A)V
$$

where $Q, K, V$ is the Query, Key, and Value corresponding to $\bar{X}$. We refer readers to Vaswani et al. (2017) for more details of the original Transformer. By only modifying the original Transformer slightly, *ms*-ESM can process residues and atoms at the same time, which makes it a versatile model for many downstream tasks. Moreover, *ms*-ESM shows great compatibility with existing pre-training model, e.g., ESM series, which allows us to bulid up a better model based on previous study more easily.

## 3 EXPERIMENTS

To verify the effectiveness of our multi-scale pre-training, we primarily evaluated the model's performance on various protein-molecule tasks(Section 3.1). In addition, to validate our model's competitive performance against other baseline models in protein-only and molecule-only tasks, we also conducted experiments on multiple protein-only tasks (Section 3.2) and molecule-only tasks (Section 3.3). For each of them, we will provide the details about fine-tuning protocol, baseline methods and performance results in corresponding paragraphs. Besides, we also perform ablation studies which discuss how different position encoding strategies (Section 3.4) effect the performance of our model. The detailed pre-training configuration, including pre-training datasets and hyperparameters, e.g., unzip ratio of residues, can be found in Appendix A.

Table 1: Performance comparison on enzyme-substrate affinity regression task.

| Method | Protein Pre-training Model | Molecule Pre-training Model | MSE↓ | $R^2$ ↑ | Pearson ↑ |
|---|---|---|---|---|---|
| Gollub et al. (2023) | / | / | / | 0.463 | 0.680 |
| Kroll et al. (2021) | / | / | 0.653 | 0.527 | 0.728 |
| XGBoost | ESM-2 35M | Uni-Mol 48M | 0.652 | 0.528 | 0.727 |
| ProSmith | ESM-2 35M | Uni-Mol 48M | 0.642 | 0.536 | 0.733 |
| XGBoost | *ms*-ESM 35M | *ms*-ESM 35M | 0.623 | 0.548 | 0.742 |
| ProSmith | *ms*-ESM 35M | *ms*-ESM 35M | **0.599** | **0.566** | **0.753** |

Table 2: Performance comparison on drug-target affinity regression task.

| Method | Protein Pre-training Model | Molecule Pre-training Model | MSE↓ | CI↑ | $r_m^2$ ↑ |
|---|---|---|---|---|---|
| Öztürk et al. (2018) | / | / | 0.261 | 0.878 | 0.630 |
| Shin et al. (2019) | / | Molecule Transformer | 0.245 | 0.887 | 0.665 |
| Nguyen et al. (2021a) | / | / | 0.229 | 0.893 | 0.685 |
| Nguyen et al. (2021b) | TAPE 38M | / | 0.228 | 0.893 | / |
| Qiu et al. (2021) | ProtBert 420M | / | 0.205 | 0.896 | 0.709 |
| Kao et al. (2021) | / | / | 0.202 | 0.907 | / |
| Yuan et al. (2022) | ESM-1b 650M | / | 0.208 | 0.913 | 0.743 |
| Yang et al. (2022) | / | / | 0.207 | 0.900 | 0.710 |
| He et al. (2023) | BiLSTM | BiLSTM | 0.196 | **0.914** | 0.744 |
| XGBoost | ESM-2 35M | Uni-Mol 48M | 0.261 | 0.885 | 0.652 |
| ProSmith | ESM-2 35M | Uni-Mol 48M | 0.219 | 0.899 | 0.711 |
| XGBoost | *ms*-ESM 35M | *ms*-ESM 35M | 0.248 | 0.889 | 0.668 |
| ProSmith | *ms*-ESM 35M | *ms*-ESM 35M | **0.191** | 0.906 | **0.759** |

## 3.1 PROTEIN-MOLECULE TASKS

**Fine-tuning Protocol** For protein-molecule tasks, we follow the benchmark protocol from Pro-Smith (Kroll et al., 2023b) to evaluate *ms*-ESM on three tasks, including enzyme-substrate affinity regression (ESAR), drug-target affinity regression , and enzyme-substrate pair classification. Specifically, each task provides the protein residue sequence and molecule SMILES string as input and asks models to tell whether the protein-molecule pair has high affinity. As our *ms*-ESM can not process SMILES strings, we first use RDKit (Landrum et al., 2013) to generate corresponding molecule conformation according to the SMILES and then extract the atom sequence and atom-scale position encoding for *ms*-ESM. For more details of the fine-tuning, see Appendix B.1.

**Baselines** We compare *ms*-ESM with multiple baselines on each tasks, including supervised and pre-training baseline. For each baseline, we list their protein pre-training model and molecule pre-training model in corresponding tables. More details of each baseline can be seen in corresponding papers. As only ProSmith (Kroll et al., 2023b) provides a framework to combine protein pre-training model and molecule pre-training model, we follow their framework by substituting both protein model and molecule model to *ms*-ESM for fair comparison. We also provide an XGBoost (Chen & Guestrin, 2016) variant of ProSmith, which takes the concatenation of protein and molecule representation as features and can directly evaluates whether two representations can work well together.

**Results** Table 1, Table 2, and Table 3 show the experiment results of *ms*-ESM and competitive baselines on three tasks. From the results, we can get the summarization as follows: (1) *ms*-ESM achieves the SOTA result on most metrics. (2) ProSmith and XGBoost based on our *ms*-ESM are always better than the version that combines two separate pre-training models. (3) *ms*-ESM can beat the methods which based on much larger pre-training models.

These phenomenons obviously indicate that **pre-training proteins and molecules in one model can further release the power of pre-training technique on protein-molecule tasks**. Fusing two separate pre-training models can be a sub-optimal for such tasks and the problem can not be fixed by using larger pre-training models.

Table 3: Performance comparison on enzyme-substrate pair classification task.

| Method | Protein Pre-training Model | Molecule Pre-training Model | ACC ↑ | MCC ↑ | ROC-AUC ↑ |
|---|---|---|---|---|---|
| Kroll et al. (2023b) | ESM-1b 650M | / | 91.5% | 0.780 | **0.956** |
| XGBoost | ESM-2 35M | Uni-Mol 48M | 89.9% | 0.729 | 0.941 |
| ProSmith | ESM-2 35M | Uni-Mol 48M | 90.8% | 0.754 | 0.943 |
| XGBoost | *ms*-ESM 35M | *ms*-ESM 35M | 90.6% | 0.750 | 0.943 |
| ProSmith | *ms*-ESM 35M | *ms*-ESM 35M | **91.8%** | **0.781** | 0.954 |

Table 4: Performance comparison on the contact prediction task.

| Method | Short Range ↑ | | | Medium Range ↑ | | | Long Range ↑ | | |
|---|---|---|---|---|---|---|---|---|---|
| | P@L | P@L/2 | P@L/5 | P@L | P@L/2 | P@L/5 | P@L | P@L/2 | P@L/5 |
| TAPE 38M | **0.28** | **0.35** | 0.46 | 0.19 | 0.25 | 0.33 | 0.17 | 0.20 | 0.25 |
| ResNet 38M | 0.25 | 0.34 | 0.46 | 0.18 | 0.25 | 0.35 | 0.10 | 0.13 | 0.17 |
| ESM-2 35M | 0.20 | 0.29 | 0.46 | 0.22 | **0.32** | **0.45** | **0.30** | **0.39** | **0.49** |
| *ms*-ESM 35M | 0.21 | 0.31 | **0.48** | **0.23** | **0.32** | **0.45** | 0.29 | 0.38 | 0.48 |

## 3.2 PROTEIN-ONLY TASKS

**Fine-tuning Protocol**  We use protein-only tasks to evaluate whether *ms*-ESM still has good understanding of proteins. Specifically, we follow TAPE (Rao et al., 2019) and use the tasks secondary structure prediction and contact prediction to judge the ability of protein pre-training models in protein structure understanding. To perform secondary structure prediction, models need to understand the local structure of proteins, e.g., helix and strand. For the task contact prediction, it requires models to have a good understanding of proteins more globally. As *ms*-ESM supports protein residue sequences, we follow TAPE's protocol strictly. For a fair comparison, we remove the test data that appears in the pre-training data, and the proportion of this part of the data is less than 4‰. For more details of the fine-tuning protocol, readers can find them in Appendix B.2.

**Baselines**  For the protein-only benchmark, we chose several popular protein pre-training models as our baselines. TAPE (Rao et al., 2019) and ResNet (Rao et al., 2019) use a Transformer (Vaswani et al., 2017) and a dilated residual network (Yu et al., 2017) as the backbone network to train a masked language model (MLM) respectively. Because *ms*-ESM loads a checkpoint from ESM-2 as the parameter initialization, we also include the ESM-2 model (Lin et al., 2023) in our comparison.

**Results**  We report the results of contact prediction and secondary structure prediction in Table 4 and Table 5 respectively. Although *ms*-ESM does not achieve the best performance among comparing methods. However, as shown in the tables, *ms*-ESM performs very similarly to ESM-2 on both secondary structure prediction and contact prediction, which indicates that **we do preserve the local and global understanding of proteins originally from ESM-2**. Promisingly, *ms*-ESM can have a better protein understanding by simply using a larger ESM-2 as the parameter initialization. We leave it as the future work.

Table 5: Performance comparison on secondary structure prediction task.

| Method | SS3(ACC) ↑ | | | SS8(ACC) ↑ | | |
|---|---|---|---|---|---|---|
| | cb513 | ts115 | casp12 | cb513 | ts115 | casp12 |
| TAPE 38M | 0.73 | 0.77 | 0.71 | 0.59 | 0.64 | 0.59 |
| ResNet 38M | 0.75 | 0.78 | 0.72 | 0.58 | 0.64 | 0.58 |
| ESM-2 35M | **0.80** | **0.82** | **0.74** | **0.65** | **0.70** | **0.61** |
| *ms*-ESM 35M | 0.79 | 0.81 | **0.74** | 0.63 | 0.69 | 0.60 |

## 3.3 MOLECULE-ONLY TASKS

Table 6: Performances on molecular property classification and regression tasks.

| Method | Reg. (MAE) ↓ | | Cls. (AUC,%) ↑ | |
|---|---|---|---|---|
| | QM8 | QM9 | HIV | MUV |
| D-MPNN | 0.0190 | 0.00814 | 77.1 | 78.6 |
| N-Gram$_{XBG}$ | 0.0215 | 0.00964 | 78.7 | 74.8 |
| GROVER$_{large}$ | 0.0224 | 0.00986 | 68.2 | 67.3 |
| MolCLR | 0.0178 | / | 78.1 | 79.6 |
| GEM | 0.0171 | 0.00746 | 80.6 | 81.7 |
| Uni-Mol$_H$ | **0.0156** | **0.00467** | **80.8** | **82.1** |
| Uni-Mol$_{w/o\ H}$ | 0.0160 | 0.00540 | 78.3 | 72.0 |
| *ms*-ESM$_{w/o\ H}$ | 0.0166 | 0.00590 | 74.9 | 72.6 |

Table 7: The ablation study on multi-scale position encoding.

| Method | ESAR | |
|---|---|---|
| | MSE ↓ | $R^2$ ↑ |
| Vanilla *ms*-ESM | 0.627 | 0.546 |
| w/o ASPE | 0.639 | 0.537 |
| w/o RSPE in atoms | 0.627 | 0.547 |

**Fine-tuning Protocol**    We use molecule-only tasks to evaluate whether we successfully make a protein pre-training model (originally trained under pure protein situations) work well under pure molecule situations. As we use the molecule data from Uni-Mol (Zhou et al., 2023) to train our *ms*-ESM , we also adopt the fine-tuning protocol of Uni-Mol to evaluate the molecule understanding ability of our models. Specifically, we only use two molecule property regression tasks (QM8, QM9) and two molecule property classification tasks (HIV, MUV) in our comparison, because each of these tasks can provide a large dataset ($> 10000$ instances), which can avoid the over-fitting problems in the fine-tuning stage and give us a more stable experiment results. For more fine-tuning details and results on more molecule-only tasks, we refer readers to Appendix B.3 and Appendix D.

**Baselines**    Following Uni-Mol, we use multiple supervised and pre-training methods as our baselines. The details of each baseline model can be found in the Uni-Mol paper (Zhou et al., 2023). Notably, according to whether hydrogen atoms are removed or not in pre-training, there are two versions of Uni-Mol, i.e., Uni-Mol$_H$ and Uni-Mol$_{w/o\ H}$. We report the results of the two versions of Uni-Mol in Table 6 for fair comparison, because we remove hydrogen atoms when training *ms*-ESM. We only distinguish the two versions of Uni-Mol here. Without further explain, we refer Uni-Mol$_{w/o\ H}$ to Uni-Mol.

**Results**    Table 6 shows the experiment results of both molecular property classification and regression tasks. Similar to protein-only tasks, *ms*-ESM  is not the best method for molecular property prediction. Nevertheless, it is comparable to the Uin-Mol (without hydrogen atoms version) on most of tasks, which makes it still a strong method for molecule-only tasks. Considering that retaining hydrogen atoms on Uni-Mol can improve performance, we believe that we can further boost the *ms*-ESM's performance by keeping hydrogen atoms in pre-training. In summary, the results on molecule-only tasks clearly demonstrate that **we successfully make a protein pre-training model works well under pure molecule situations**.

### 3.4    ABLATION

**Multi-scale Position Encoding**    To validate the effectiveness of multi-scale position Encoding, we conduct ablation tests under two conditions: one without using atom scale PE (ASPE) and another without providing residue scale PE (RSPE) to atoms. The task employed is the enzyme-substrate affinity regression task. As shown in Table 7, when atom scale PE is not used, the model's performance suffered significantly, which is because the model fails to capture positional information of atoms without providing atom scale PE. On the other hand, when residue scale PE is not provided to atoms, the model's performance remains nearly unchanged. This suggests that for atom-scale information, 3D structural information is more crucial, and since the mapping relationship from residues to atoms is straightforward, there may be no need to provide residue scale PE to atoms to distinguish their corresponding residues.

### 3.5    VISUALIZATION

To provide a more intuitive illustration of the higher consistency in protein and small molecule representations learned by this multi-scale unified model *ms*-ESM, we conducted visual comparisons

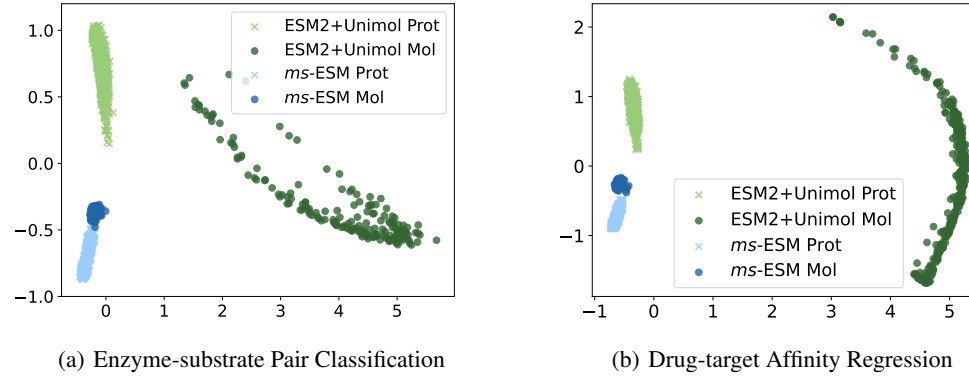

(a) Enzyme-substrate Pair Classification     (b) Drug-target Affinity Regression

Figure 4: Visualization of representations learned by *ms*-ESM and ESM2 + Unimol.

between proteins' and molecules' features extracted by *ms*-ESM and the proteins' features extracted by ESM2 along with the molecules' features extracted by Unimol in both the enzyme-substrate pair classification task and drug-target affinity regression task. As depicted in Figure 4, the proteins' and molecules' representations learned by *ms*-ESM model are closer. This implies that *ms*-ESM can construct a more unified semantic representation for both proteins and molecules data.

## 4 RELATED WORK

**Protein Pre-training**   Pre-training has been proved to be an efficient technique in many domains, like natural language processing and protein engineering. Existing work studies protein pre-training mainly in two ways: (1) Sequence-based methods learn protein primary sequences to capture the biochemical and co-evolutionary knowledge. ESM series models (Rives et al., 2021; Lin et al., 2022b; 2023) use vanilla masked language modeling to learn protein representations on evolutionary scale. Aiming at the specific contact prediction task, Rao et al. (2021) further extends the masked language modeling to multiple sequence alignment (MSA) data. Inspired by the large language model (LLM), ProtGPT2 (Ferruz et al., 2022), ProGen(Madani et al., 2023), and ProGen2 (Nijkamp et al., 2022) scale up the model size of protein language model and show promising results in protein generation tasks. (2) Structure-based methods directly learn protein structure in different levels. Gligorijević et al. (2021); Zhang et al. (2022); Xu et al. (2022) learn residues from a local part of protein structures. Jing et al. (2020); Zhang et al. (2023) try to capture atomic structure knowledge in proteins. We develop *ms*-ESM based on ESM. Differently, *ms*-ESM is a mixture of sequence and structure-based methods, which gives it the ability to process information from different scales and makes it a versatile model.

**Unified Molecular Modeling**   Because of the huge scale difference of proteins and small molecules, it is challenging to model both of them in a unified style. As far as we know, Uni-Mol (Zhou et al., 2023) is the only method that tries to process proteins and molecules uniformly. Uni-Mol realizes the uniformity by directly modeling proteins and molecules at atom scale. However, because an entire protein contains hundreds of thousands of atoms, Uni-Mol can only model a local structure of proteins, i.e., protein pocket. Unlike Uni-Mol, as *ms*-ESM only unzips partial residues into their corresponding atoms, it can handle an entire protein efficiently. We also provide some discussions of molecular modeling in Appendix C.

## 5 CONCLUSIONS

In this study, we propose a multi-scale protein language model *ms*-ESM, which realizes multi-scale unified molecular modeling by pre-training on multi-scale code-switch protein sequence and describing relationships among residues and atoms with a multi-scale position encoding. Experiment results show that *ms*-ESM outperforms previous methods in protein-molecule tasks and is on par with the state-of-the-art in protein-only and molecule-only tasks.

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

## A  PRE-TRAINING CONFIGURATION

**Pre-training Datasets**  We use a mixture of protein dataset and molecule dataset as the pre-training dataset. As Euclidean distance is required by atom scale position encoding, we use datasets come with the structure information, i.e., atom coordinates.

For the protein dataset, we use AlphaFold DB (Varadi et al., 2022) dataset, which contains 3M protein sequences and structures predicted by AlphaFold2 (Jumper et al., 2021) with high confident.

For the molecule dataset, we use the dataset provided by Zhou et al. (2023), which contains 19M molecules and 209M conformations generated by ETKGD (Riniker & Landrum, 2015) and Merck Molecular Force Field (Halgren, 1996).

Unlike Zhou et al. (2023), we do not train two models using two datasets respectively, instead we mix these two datasets and only train one *ms*-ESM.

**Hyperparameters** We implement *ms*-ESM with 12 stacked 20-head Transformer layers mentioned in Section 2.4. The model dimension and feedforward dimension of each Transformer layer are 480 and 1920. The total number of *ms*-ESM's parameters is 35M. We use Adam (Kingma & Ba, 2014) and polynomial learning rate scheduler to train *ms*-ESM and set the learning rate 4e-4, weight decay 1e-2, warmup step 5000. The total training step is 300K and each batch has 256K tokens at maximum. We train *ms*-ESM on 16 NVIDIA A100 GPU cards for 3 days. *ms*-ESM is compatible with ESM series, so we load a ESM 35M checkpoint as the initialization of *ms*-ESM. For ORDER procedure, we use the default order in PDB (protein) and SDF (molecule) files as the order assigned to the atom set. To elaborate, PDB and SDF serve as standard formats for describing atomic structures of proteins and small molecules, respectively. In both formats, atoms follow specific sorting principles. In our study, we directly utilize the sorted atoms for ease of implementation. It is important to note that, given our atom-scale position encoding employs Euclidean distance to describe positional relationships, the permutation of atom order does not impact our pre-training model. When pre-training, 6.5% of residues are unzipped as the main experimental setting. For more pre-training hyperparameters see Table 8.

Table 8: *ms*-ESM  hyperparameters for pre-training.

| hyperparameters | Value |
|---|---|
| Learning rate | 4e-4 |
| LR scheduler | polynomial_decay |
| End learning rate | 4e-5 |
| Warmup updates | 5000 |
| Max update | 300000 |
| Max tokens | 262144 |
| Distance loss function and its weight | Smooth L1, 10.0 |
| MLM loss function and its weight | Cross entropy, 4.0 |
| Dropout | 0.0 |
| Attention dropout | 0.0 |
| Activation dropout | 0.0 |
| Num of encoder layers | 12 |
| Num of encoder attention heads | 20 |
| Encoder embedding dim | 480 |
| Encoder feedForward dim | 1920 |
| Adam $(\beta_1, \beta_2)$ | (0.9,0.98) |
| Mask ratio | 0.15 |
| Unzip ratio | 0.065 |
| Distance noise $\epsilon$ | 1 Å |

## B  FINE-TUNING DETAILS

Here, we provide more implementation details in fine-tuning downstream tasks. We also provide the statistics of each fine-tuning dataset in Table 9.

### B.1  FINE-TUNING DETAILS OF PROTEIN-MOLECULE TASKS

**Fine-tuning Datasets** Following ProSmith (Kroll et al., 2023b), we finetune *ms*-ESM and all baseline models on dataset KM (Kroll et al., 2021), Davis (Davis et al., 2011), and ESP (Kroll et al., 2023a) for enzyme-substrate affinity regression, drug-target affinity regression, and enzyme-substrate pair classification respectively. The KM dataset contains experimental affinity constants of

11,676 enzyme-substrate pairs. The Davis dataset provides 30,056 binding affinities for pairs of 72 drugs and 442 proteins. The ESP dataset consists of 68,754 positive/negative enzyme-substrate pairs with experimental evidence. We use the standard data split provided by ProSmith in fine-tuning.

**Fine-tuning Framework**    As mentioned in section 3.1, we use ProSmith's framework for a fair comparison. Specifically, the framework contains three main modules, i.e., molecule encoder, protein encoder, and fusion block. Two encoders extract features from proteins and molecules severally. The fusion block is a Transformer model, which is responsible for fusing protein and molecule features. The fused features are further used to regress the affinity values or predict binary affinity. We apply our model to ProSmith's framework by replacing both protein and molecule encoder with *ms*-ESM. We also provide the results of an XGBoost (Chen & Guestrin, 2016) variant of ProSmith, which removes the fusion block and uses simple concatenation for feature fusing. Note that we freeze both encoders in the experiments as suggested by ProSmith.

**Fine-tuning Hyperparameters**    We directly use the hyperparameters provided by ProSmith. Specifically, the fusion block for three tasks has 6 layers of Transformer whose hidden size is 768. The epoch number is 100 and the learning rate is 1e-5. The batch sizes of the three tasks are 12, 12, and 24. We use Adam (Kingma & Ba, 2014) as the optimizer for ProSmith and GBDT (Ke et al., 2017) with 500 iterations as the predictors for XGBoost.

### B.2    Fine-tuning details of protein-only tasks

**Fine-tuning Datasets**    Following TAPE's protocol (Rao et al., 2019), we evaluate *ms*-ESM on secondary structure prediction and contact prediction tasks. Specifically, for secondary structure prediction, we use data from Klausen et al. (2019) as training and validation sets and use CB513 (Cuff & Barton, 1999), CASP12 (Moult et al., 2018), and TS115 (Yang et al., 2018) as test sets. The training and validation sets are filtered at the 25% sequence identity threshold with these test tests. The final training, validation and three test sets have 8678, 2170, 513, 21, 115 protein sequences, respectively. For contact prediction tasks, we use training, validation, and test sets from ProteinNet (AlQuraishi, 2019) with training and validation sets filtered at the 30% sequence identity threshold. The final training, validation, and test sets have 20, 24, 13945 protein sequences.

**Fine-tuning Framework**    As suggested by TAPE, for both protein-only tasks, we use *ms*-ESM as the protein encoder. When doing secondary structure prediction, we use a linear output layer to predict the secondary structure each residue belongs to. When handling the contact prediction task, we use the attention from the last layer as features and then use a linear layer to predict whether these two residues have contact or not. Notably, both input of these two tasks is only protein sequences without structural information. Therefore, when using *ms*-ESM to handle these two tasks, we turn off the unzip.

**Fine-tuning Hyperparameters**    We set up all the hyperparameters aligned to TAPE. For secondary structure prediction, the epoch is 5, 5000, batch size is 10, and learning rate is 0.001. For contact prediction, the epoch is 5, batch size 64, and learning rate is 3e-5. We use AdamW (Loshchilov & Hutter, 2017) as the optimizer in secondary structure prediction and Adam (Kingma & Ba, 2014) in contact prediction.

### B.3    Fine-tuning details of molecule-only tasks

**Fine-tuning Datasets**    We use the fine-tuning data of Uni-Mol (Zhou et al., 2023) to evaluate the molecule understanding ability of *ms*-ESM. Specifically, we use QM8, QM9 datasets for molecular property regression and HIV, MUV datasets for molecular property classification, which have 21786, 133885, 41127, 93087 molecules, respectively. The data split is also provided by Uni-Mol.

**Fine-tuning Framework**    Following Uni-Mol, a special token, i.e., `[CLS]`, also exists in *ms*-ESM. Similar to NLP/CV, we simply use the representation of `[CLS]` to represent the whole molecule, and then use a linear head for fine-tuning on downstream tasks. For each molecule, we use the 3D conformation provided by Zhou et al. (2023) as the input of *ms*-ESM. In the fine-tuning stage, we do not add noises to atom coordinates.

**Fine-tuning Hyperparameters**    For a fair comparison, we did not search the best hyperparameters. Instead, we set up all the hyperparameters aligned to Uni-Mol. Specifically, the batch sizes for these four tasks are 32, 128, 256, and 128. The learning rates are 1e-4, 1e-4, 5e-5, and 2e-5. The training epochs are 40, 40, 5, and 40. We use Adam optimizer for all the tasks.

## C   MORE RELATED WORK

**Molecular Modeling**    Regarding the modality of molecules, studies on molecular modeling can be categorized into three groups. (1) 1D-based methods: These represent molecules with SMILES strings and employ language modeling techniques, such as masking and contrastive self-supervision, to enhance molecular representation (Wang et al., 2019; Honda et al., 2019; Chithrananda et al., 2020; Zhang et al., 2021a; Xue et al., 2020; Guo et al., 2022). (2) 2D-based methods: These represent molecules with molecular graphs, sharing common ideas with general graph modeling. Some methods (Rong et al., 2020; Li et al., 2020; Zhang et al., 2021b; Li et al., 2021) mask key substructures of molecular graphs, like motifs and functional groups, and task models with reconstructing the masked parts. Others (Wang et al., 2022; Fang et al., 2022c; Lin et al., 2022a) align views from positive pairs (corrupt versions of the same graph) and simultaneously contrast views from negative pairs (different graphs). (3) 3D-based methods: These directly utilize the 3D structure of molecules, aligning closely with our work. Earlier studies incorporated 3D information as an auxiliary input for 2D-based methods (Liu et al., 2021; Li et al., 2022b; Zhu et al., 2022; Stärk et al., 2022). More recent methods focus on molecular modeling with pure 3D inputs (Fang et al., 2022a; Zhou et al., 2023; Luo et al., 2022; Zaidi et al., 2022; Liu et al., 2022; Jiao et al., 2023). Three self-supervised techniques have been designed: geometry masking, geometry predicting, and denoising. For masking, Fang et al. (2022a) mask bond information, while Zhou et al. (2023) mask atom types, requiring models to predict masked information based on remaining context. For predicting, Fang et al. (2022a) proposes an atomic prediction task with bond information to capture global structure from local information. For denoising, models reconstruct 3D structures by adjusting corrupted structures. When corrupting structures, Zhou et al. (2023); Luo et al. (2022); Zaidi et al. (2022) add Gaussian noise to each atom of the input molecule. Several methods further introduce E(3)- and SE(3)-invariance inductive bias to the denoising technique (Zhou et al., 2023; Liu et al., 2022; Jiao et al., 2023).

## D   RESULTS ON MORE MOLECULE-ONLY TASKS

To verify that the proposed *ms*-ESM does not suffer from significant performance drops on molecular modeling tasks, we select more molecular tasks from Uni-Mol(Zhou et al., 2023) to test model performance. On all tasks, our model uses the same fine-tuning and evaluation protocols with Uni-Mol(Zhou et al., 2023). As shown in 10, *ms*-ESM has a similar performance with Uni-Mol$_{w/o H}$. Even on some tasks (such as BACE, SIDER and MUV), *ms*-ESM  has stronger performance compared to Uni-Mol$_{w/o H}$, which indicates that the two model have similar molecular modeling ability.

## E   MORE RESULTS ON ABLATION STUDY

### E.1   ABLATION ON PRE-TRAINING OBJECTIVES

Significantly decreased model performance is observed when either the masked atom type prediction loss or the pair-wise distance recovery loss is omitted, as shown in Table 11. Notably, removing the pair-wise distance recovery loss results in a greater performance loss compared to omitting the masked atom type prediction loss. For molecular representation learning, the training loss of *ms*-ESM  on the atomic scale is only masked atom type prediction loss without using pair-wise distance recovery loss. But more than half of the atoms in a molecule are carbon atoms, which makes the model learn very little information on the atomic scale using only masked atom type prediction. And *ms*-ESM  will also not be able to learn structural information from the unzipped residues without using pair-wise distance recovery loss. These results indicate that, while both atom type and structural information are vital for atomic-scale details, structural information holds greater importance.

Table 9: The statistics of downstream datasets in one table. ESAR: Enzyme-Substrate Affinity Regression, DTAR: Drug-Target Affinity Regression, ESPC: Enzyme-Substrate Pair Classification, SSP: Secondary Structure Prediction, CP: Contact Prediction, MPR: Molecular Property Regression, MPC: Molecular Property Classification.

| Task | Protein-Molecule Task | | | Protein-Only Task | | Molecule-Only Task | | | |
| | ESAR | DTAR | ESPC | SSP | CP | MPR | | MPC | |
| Dataset | KM | Davis | ESP | NetSurfP-2.0, CB513[3] CASP12, TS115 | ProteinNet | QM8 | QM9 | HIV | MUV |
|---|---|---|---|---|---|---|---|---|---|
| Train | 8407 | 24045 | 49876 | 8678 | 20 | 17428 | 107108 | 32901 | 74469 |
| Valid | 934 | 3006 | 5540 | 2170 | 24 | 2179 | 13388 | 4113 | 9309 |
| Test | 2335 | 3005 | 13336 | 513/21/115 | 13945 | 2179 | 13389 | 4113 | 9309 |
| Total | 11676 | 30056 | 68754 | 11497 | 13989 | 21786 | 133885 | 41127 | 93087 |

Table 10: More experimental results on molecular tasks. Details of baselines can be found in Uni-Mol(Zhou et al., 2023).

| Method | Reg. (MAE) ↓ | | | Cls. (AUC,%) ↑ | | | | | | |
|---|---|---|---|---|---|---|---|---|---|---|
| | QM7 | QM8 | QM9 | BACE | BBBP | TOX21 | PCBA | SIDER | HIV | MUV |
| D-MPNN | 103.5 | 0.0190 | 0.00814 | 80.9 | 71.0 | 75.9 | 86.2 | 57.0 | 77.1 | 78.6 |
| Attentive FP | 72.0 | 0.0179 | 0.00812 | 78.4 | 64.3 | 76.1 | 80.1 | 60.6 | 75.7 | 76.6 |
| N-Gram$_{RF}$ | 92.8 | 0.0236 | 0.01037 | 77.9 | 69.7 | 74.3 | - | 66.8 | 77.2 | 76.9 |
| N-Gram$_{XBG}$ | 81.9 | 0.0215 | 0.00964 | 79.1 | 69.1 | 75.8 | - | 65.5 | 78.7 | 74.8 |
| GROVER$_{base}$ | 94.5 | 0.0218 | 0.00984 | 82.6 | 70.0 | 74.3 | 76.5 | 64.8 | 62.5 | 67.3 |
| GROVER$_{large}$ | 92.0 | 0.0224 | 0.00986 | 81.0 | 69.5 | 73.5 | 83.0 | 65.4 | 68.2 | 67.3 |
| PretrainGNN | 113.2 | 0.0200 | 0.00922 | 84.5 | 68.7 | 78.1 | 86.0 | 62.7 | 79.9 | 81.3 |
| GraphMVP | - | - | - | 81.2 | 72.4 | 75.9 | - | 63.9 | 77.0 | 77.7 |
| MolCLR | 66.8 | 0.0178 | - | 82.4 | 72.2 | 75.0 | - | 58.9 | 78.1 | 79.6 |
| GEM | 58.9 | 0.0171 | 0.00746 | 85.6 | 72.4 | 78.1 | 86.6 | **67.2** | 80.6 | 81.7 |
| Uni-Mol$_H$ | **41.8** | **0.0156** | **0.00467** | **85.7** | **72.9** | **79.6** | **88.5** | 65.9 | **80.8** | **82.1** |
| Uni-Mol$_{w/o\ H}$ | 58.9 | 0.0160 | 0.00540 | 83.2 | 71.52 | 78.92 | 88.12 | 57.71 | 78.3 | 72.0 |
| *ms*-ESM$_{w/o\ H}$ | 60.9 | 0.0166 | 0.00590 | 83.52 | 67.41 | 75.39 | 86.15 | 63.59 | 74.9 | 72.6 |

Table 11: More experimental results on ablation study.

| Method | ESAR | |
|---|---|---|
| | MSE ↓ | $R^2$ ↑ |
| Vanilla *ms*-ESM | 0.627 | 0.546 |
| w/o ASPE | 0.639(+0.012) | 0.537(-0.009) |
| w/o RSPE in Atoms | 0.627(+0.0) | 0.547(+0.001) |
| w/o Masked Atom Type Loss | 0.642(+0.015) | 0.535(-0.011) |
| w/o Pair-wise Distance Recovery Loss | 0.645 (+0.018) | 0.533(-0.013) |
| w/o Molecular Data | 0.648 (+0.021) | 0.531(-0.015) |
| w/o Protein Data | 0.708(+0.081) | 0.487(-0.059) |
| w/o Unzip Operation | 0.638 (+0.011) | 0.538(-0.008) |

## E.2    ABLATION STUDY ON PRE-TRAINING DATA

A significant decrease in model performance is observed when either molecular data or protein data is excluded, as shown in Table 11. Interestingly, the removal of protein data leads to a more substantial performance decline than omitting molecular data, which indicates that when the model is not trained with protein data, it quickly forgets protein-related knowledge, resulting in a notable decline in overall performance. But the model can still learn some atomic scale information through unzip operations without molecular data. This explains why the model performs better without molecular data compared to the scenario without protein data.

