# OpenReview forum: "Multi-Scale Protein Language Model for Unified Molecular Modeling"
_ICLR.cc/2024/Conference — Submitted to ICLR 2024_

### Official Review · Reviewer_54HD · 2023-10-20

**Soundness:** 1 poor
**Presentation:** 3 good
**Contribution:** 1 poor
**Rating:** 3
**Confidence:** 5

**Summary:**

The paper extends the concept of protein language models to atom-scale and molecular data. Methodologically, the contribution of the paper are three-fold: (1) the development of a universal transformer tailored for molecule and protein data; (2) the introduction of a code-switch protein sequence approach to unpack residues; (3) the presentation of a multi-scale position encoding designed specifically for code-switch sequences. The authors claim that they effectively demonstrate the efficacy of their approach through enzyme-substrate and drug-target affinity tasks. Additionally, their method achieves comparable results in areas such as contact prediction, secondary structure prediction, and molecular property prediction tasks.

**Strengths:**

1. The idea of designing a universal transformer for molecules and proteins are of high potential. Such an approach has the potential to unify various downstream tasks related to both molecules and proteins. However, there are evident flaws in the methodology presented in the paper.
2. To my knowledge, the related work section about protein pre-training is thorough and comprehensive.

**Weaknesses:**

1. There are significant flaws in the design of code-switch protein sequences, which can potentially make the model learn meaningful insights from the unzipped sequences.
2. The concept of multi-scale position encoding essentially merges residue- and atom-level embeddings. This has been previously proposed in other papers and, thus, lacks novelty.
3. The results present in the experimental section only shows marginal improvements over the established baselines. Additionally, the unified pre-training appears to diminish performance on tasks focused solely on proteins or molecules. This compromises its viability for practical applications.
4. The related work section seems to overlook several pivotal studies pertinent to molecular modeling.

For details, please refer to the Question section.

**Questions:**

1. The concept of unzipping residues into atom sequences and predicting the masked atom type appears flawed. Given the residue type or the types of adjacent atoms, deducing the type of the masked atom becomes straightforward, since the atom set is predetermined for each residue type. This indicates that unzipping residues does not introduce any unique or non-trivial information. Additionally, there is a lack of experiments to demonstrate that the masked atom type prediction loss contributes any meaningful insight.
2. As shown in the Tables 2 and 3, ProSmith's results with ms-ESM offer only a slight enhancement over baselines, such as He et al. (2023) in Table 2 and Kroll et al. (2023b) in Table 3.
3. Are the pre-trained language models fine-tuned for specific tasks? If not, the tables should encompass results with fine-tuning.
4. The protein-only tasks seem trivial when protein structures are used as inputs. With knowledge of protein tertiary structures, it becomes easy to determine if two residues are in contact and to identify the secondary structure. Moreover, since the pre-training task includes pairwise distance prediction, there's potential for data leakage. The test data might overlap with the pre-training dataset.
5. Despite the possible data and information leakage in the experimental framework, the proposed method fails to outperform standard protein language models like ESM-2 in tasks such as contact and secondary structure prediction. Given that the model incorporates a broader range of pre-training loss than ESM-2, these results suggest that universal pre-training across both proteins and molecules may not offer advantages in protein-only or molecule-only tasks. This significantly weakens the paper's primary claim: the benefit of combining protein and molecule data for pre-training.
6. The ablation study only consider ablations on position encoding, neglecting the paper's other two significant contributions. It would be beneficial for the authors to explore comparisons with protein- or molecule-only pre-training, consider pre-training without unzipped sequences, and evaluate the impact of removing each pre-training loss.

Overall, I recognize the significance of developing universal models for molecule and protein pre-training. However, the paper exhibits considerable flaws in its methodologies and experimental sections, making it below the standard expected for publication.

---

> ### Author Response · Authors · 2023-11-18
> **Response to Reviewer 54HD (part 1)**
>
> We express our gratitude to Reviewer 54HD for the valuable suggestions. In the following parts, we will thoroughly address your concerns regarding the design of code-switch sequences, technical novelty, model performance, ablation study, and other crucial aspects. Additionally, we will provide a detailed explanation to clarify any misunderstanding about the input of ms-ESM in downstream tasks. These updates have been incorporated into the revised manuscript, aiming to enhance the clarity of our paper. We welcome any further comments or feedback you may have.
>
> **Q1**: Given the residue type or the types of adjacent atoms, deducing the type of the masked atom becomes straightforward. There are significant flaws in the design of code-switch protein sequences, which can(not) potentially make the model learn meaningful insights from the unzipped sequences.
>
> **A1**:  Thanks for your comments. However, it seems that **the concluded significant flaws of code-switch protein sequence design could be caused by some misunderstandings.** The code-switch design and the unzip operation play crucial roles in multi-scale molecular modeling. Sorry for the confusion and we clarify it as follows:
> - **Unzipping residues also introduces non-trivial structural information for the model, aiding ms-ESM in learning diverse structural knowledge of residues.** Despite the limited types of amino acids and the fixed types of atoms within a given residue, structure-based training (pair-wise distance recovery) is conducted within an unzipped residue at the atomic scale. The structure of a single residue is thus highly diverse. Therefore, unzipping residues enables the model to acquire diverse structural information at the residue level.
> - **The unzip operation allows protein language models to learn associations between proteins and molecules and reuse the knowledge acquired by the model at the residue scale.**
>   - Similar to existing work [24] in cross-lingual pre-training, MLM training on code-switch sequences enables the model to capture correlations between information from the atom and residue scales.
>   - By retaining information at the original residue scale, the model can fully leverage the knowledge embedded in protein language models.
>   - The number of residue types does not impact the establishment of this association, because atom-scale information contains diverse structural details.
> - **Evidence from experimental results and related works**:
>   - Ablation experiments demonstrate the effectiveness of the unzip operation.
>     - Supporting the effectiveness of this approach, ablation experiments (as shown in Table D) reveal a significant performance decrease in protein-molecule tasks when the unzip operation is omitted. This underscores the crucial significance of this approach in constructing the semantic association between the two types of information.
>   - Related work also mentions the effectiveness of this approach.
>     - Concurrent work titled "Generalized Biomolecular Modeling and Design with RoseTTAFold All-Atom" [25] (published on bioRxiv after our submission to ICLR) introduces a similar concept called "atomization" and notes that this approach can assist the network in learning about general atomic interactions.
> In summary, both experimental results and related work affirm the effectiveness of this approach, forming the foundational basis for constructing a unified semantic representation across multiple scales.
>
> **Q2**: The concept of multi-scale position encoding essentially merges residue- and atom-level embeddings. This has been previously proposed in other papers and, thus, lacks novelty.
>
> **A2**: Please check Q2 in the general response for a more in-depth discussion. In essence, our work stands out for three key reasons: i) Our core contribution lies in developing a model (not exisiting techniques) capable of uniformly processing proteins and molecules. ii) Unified molecular modeling represents a cutting-edge frontier in recent AI biological applications. iii) Integrating existing techniques into our multi-scale unified molecular modeling is a non-trivial task.
>
> **Q3**: The results present in the experimental section only show marginal improvements over the established baselines. The unified pre-training appears to diminish performance on tasks focused solely on proteins or molecules.
>
> **A3**: Please refer to the Q1 in the general response for more discussion. Briefly, i) as already mentioned in the answer to Q1 in the general response, this paper focuses on unified multi-scale modeling, aiming to obtain better results on protein-molecule tasks  instead of improving molecule/protein tasks universally. ii) We report molecule-only and protein-only results to verify that ms-ESM does not suffer from significant performance drops on separated molecular and protein modeling tasks.

---

> ### Author Response · Authors · 2023-11-18
> **Response to Reviewer 54HD (part 2)**
>
> **Q4**: As shown in the Paper Tables 2 and 3, ProSmith's results with ms-ESM offer only a slight enhancement over baselines.
>
> **A4**: Thanks for your comments. ms-ESM outperforms ESM-2+Uni-Mol in all protein-molecule tasks, and also surpasses a single strong protein model (such as ESM-1b, ProtBert) in most tasks. We will provide an analysis of the reasons from two different perspectives.
> - **Comparisons with separated protein and molecule models (ESM2 + Uni-Mol)**: ms-ESM outperforms ESM-2+Uni-Mol in all protein-molecule tasks.
>   - Our protein representation learning ability is similar to ESM-2, and our molecular representation learning ability is similar to Uni-Mol, but our performance on protrein-molecule joint modeling tasks is stronger than ESM-2 + Uni-Mol.
>     - During pre-training, we don't utilize any protein-molecule pair data just like all baselines and adopt the same fine-tuning setting. So we believe that the comparison is very fair.
>   - It is worth noting that our model maintains an advantage over ESM-2+Uni-Mol in all Protein-Molecule task metrics, which illustrates the rationale behind our unified modeling approach.
> - **Comparisons with a single strong protein model (such as ESM-1b, ProtBert)**:  Small ms-ESM can even outperform large models.
>   - Under unified modeling, ms-ESM's performance can match or surpass some single representation learning models with significantly larger parameter scales (such as ESM-1b 650M) even at a much smaller scale (35M) .
>   - This illustrates that, to enhance a model's performance in protein-molecule tasks, the most crucial strategy is to construct a unified model to obtain a consistent semantic representation. This proves the significance of the unified model.
>
> **Q5**: The related work section seems to overlook several pivotal studies pertinent to molecular modeling.
>
> **A5**:
> Thank you for your information. Previously, as we focused more on protein and unified modeling, we did not include discussions of molecular modeling in our manuscript. Subsequently, we've included relevant discussions in the Appendix. We are open to further discussing any papers that reviewers consider pertinent to our work.
> - *Molecular Modeling.* Regarding the modality of molecules, studies on molecular modeling can be categorized into three groups:
>   - 1D-based methods: These represent molecules with SMILES strings and employ language modeling techniques, such as masking and contrastive self-supervision, to enhance molecular representation [1, 2, 3, 4, 5, 6].
>   - 2D-based methods: These represent molecules with molecular graphs, sharing common ideas with general graph modeling. Some methods [7, 8, 9, 10] mask key substructures of molecular graphs, like motifs and functional groups, and task models with reconstructing the masked parts. Others [11, 12, 13] align views from positive pairs (corrupt versions of the same graph) and simultaneously contrast views from negative pairs (different graphs).
>   - 3D-based methods: These directly utilize the 3D structure of molecules, aligning closely with our work. Earlier studies incorporated 3D information as an auxiliary input for 2D-based methods [14, 15, 16, 17]. More recent methods focus on molecular modeling with pure 3D inputs [18, 19, 20, 21, 22, 23]. Three self-supervised techniques have been designed: geometry masking, geometry predicting, and denoising. For masking, [18] masks bond information, while [19] masks atom types, requiring models to predict masked information based on remaining context. For predicting, [18] proposes an atomic prediction task with bond information to capture global structure from local information. For denoising, models reconstruct 3D structures by adjusting corrupted structures. When corrupting structures, [19, 20, 21] add Gaussian noise to each atom of the input molecule. Several methods further introduce E(3)- and SE(3)-invariance inductive bias to the denoising technique [19, 22, 23].
>
> **Q6**: The protein-only tasks seem trivial when protein structures are used as inputs.
>
> **A6**: We do not consider protein-only tasks as trivial because, in these tasks, ms-ESM only processes protein sequences at the residue level, as explained in Section 3.2: "As ms-ESM supports protein residue sequences, we strictly adhere to TAPE’s protocol." In both protein-only and protein-molecule tasks, we do not supply any protein structural information to ms-ESM, which is exactly the same situation as ESM-2. Therefore, we believe the comparison is fair.
>
> **Q7**: Are the pre-trained language models fine-tuned for specific tasks? If not,the tables should encompass results with fine-tuning.
>
> **A7**: Both ms-ESM and baseline methods are fine-tuned for specific tasks. We take the pre-train and fine-tune style to evaluate ms-ESM in protein-molecule, protein-only, and molecule-only tasks. We provide the fine-tuning protocol for each of tasks in section 3.1, 3.2 and 3.3. Sorry for the confusion. We have made it more clear in our revision.

---

> ### Author Response · Authors · 2023-11-18
> **Response to Reviewer 54HD (part 3)**
>
> **Q8**: Since the pre-training task includes pairwise distance prediction, there's potential for data leakage. The test data might overlap with the pre-training dataset.
>
> **A8**:
> - Thanks for your suggestions. We have counted the overlapping data in the test data and pre-training data and re-evaluated our model. After statistics, we find that the proportion of overlapped data is extremely low (less than 4‰). And **after removing all the overlap, we retest the model and the performance remains unchanged**. We have updated the above results in our revised manuscript.
> - Statistics and re-evaluation details: We count the number of samples contained in the pre-training dataset (AlphaFold DB) in the test sets for contact prediction and secondary structure prediction tasks. The results are as follows:
>   - It can be found in Table A that the proportion of data with leakage problems is extremely small.
>   - After removing leaked data, we reevaluate the model's performance on both tasks, and the results are shown in Table B and Table C.
>   - It can be seen that after removing leaked data, the performance of ms-ESM still reamins no change.
> In summary, there is little issue of leakage during both the model training process and the downstream task testing process.
>
> Table A: The number of leaked data. CP: Contact Prediction, SSP: Secondary Structure Prediction
> |  | CP train |  CP valid | CP test | CP total | SSP train | SSP valid | SSP test casp12/cb513/ts115 | SSP total |
> | --- | --- | --- | --- | --- | --- | --- | --- | --- |
> | Original Size | 20 | 24 | 13945 | 13989 | 8678 | 2170 | 21/513/115 | 11497 |
> | Leaked Data | 0 | 0 | 12 (0.86‰) | 12(0.858‰) | 28(3.2‰) | 8(3.7‰) | 0/3(5.8‰)/0 | 39(3.4‰) |
>
> Table B: New experimental results on contact map prediction. SR: Short Range, MR: Medium Range, LR: Long Range
> |   |  SR P@L|  SR P@L/2 | SR P@L/5 | MR P@L | MR P@L/2 |  MR P@L/5 | LR P@L | LR P@L/2 | LR P@L/5 |
> | --- | --- | --- | --- | --- | --- | --- | --- | --- | --- |
> | ESM2 35M | 0.197 | 0.293 | 0.458 | 0.223 | 0.318 | 0.452 | **0.297** | **0.385** | **0.492** |
> | Origin Results in the paper  | **0.21** | **0.31** | **0.48**  | **0.23** | **0.32** | **0.45** | 0.29 | 0.38  | 0.48 |
> | Results after removing leaked data | **0.21** | **0.31** | **0.48**  | **0.23** | **0.32**  | **0.45**  | 0.29 | 0.38 | 0.48 |
>
> Table C: New experimental results on secondary structure prediction.
> |  | SS3 cb513 |SS3 ts115 |SS3 casp12 |SS8 cb513 |SS8 ts115 |SS8 casp12 |
> | --- | --- | --- | --- | --- | --- | --- |
> | ESM2 35M | **0.80** | **0.82** | **0.74** | **0.65** | **0.70** | **0.61** |
> | Origin Results in the paper | 0.79 | 0.81 | **0.74** | 0.63 | 0.69 | 0.60 |
> | Results after removing leaked data | 0.79 | 0.81  | **0.74** | 0.63 | 0.69 | 0.60  |
>
> **Q9**: Despite the possible data and information leakage in the experimental framework, the proposed method fails to outperform standard protein language models like ESM-2 in tasks such as contact and secondary structure prediction. This significantly weakens the paper's primary claim: the benefit of combining protein and molecule data for pre-training.
>
> **A9**:
> - About the leakage problem:
>   - Information leakage:  The input of ms-ESM in protein-only tasks only contains protein sequences at the residue level, thus there is no information leakage in these tasks. For more details, please refer to Q6.
>   - Data leakage: There is almost no leakage of test data in protein-only tasks, and we also reevaluate the model's performance and find that the model performance remains almost unchanged. For more details, please refer to Q8.
> - The proposed method fails to outperform ESM-2:
>   - For more details, please refer to Q1 in the general response.
>     - ms-ESM retains comparable ability to ESM-2 in protein-only tasks.
>     - Even though ms-ESM can't beat ESM-2 in all protein-only tasks and ms-ESM can't beat Uni-Mol in all molecule-only tasks, ms-ESM still can beat  ESM-2+Uni-Mol in ALL protein-molecule tasks when using the same fine-tuning and evaluating protocol. This demonstrates the effectiveness of unified modeling.

---

> ### Author Response · Authors · 2023-11-18
> **Response to Reviewer 54HD (part 4)**
>
> **Q10**: The ablation study only considers ablations on position encoding, neglecting the paper's other two significant contributions. It would be beneficial for the authors to explore comparisons with protein- or molecule-only pre-training, consider pre-training without unzipped sequences, and evaluate the impact of removing each pre-training loss
>
> **A10**: Thank you very much for your helpful questions. We have added more ablation experiments (shown in Table D) and also added them to our revised manuscript. Here is the analysis of these results.
> - Ablation Study on Pre-training Objectives:
>   - Significantly decreased model performance is observed when either the masked atom type prediction loss or the pair-wise distance recovery loss is omitted.
>   - Notably, removing the pair-wise distance recovery loss results in a greater performance loss compared to omitting the masked atom type prediction loss.
>     - Explanation:
>       - Without using pair-wise distance recovery loss, the training loss of ms-ESM on the atomic scale is only masked atom type prediction loss. But more than half of the atoms in a molecule are carbon atoms, which makes the model learn very little information on the atomic scale using only masked atom type prediction.
>       - Without the pair-wise distance recovery loss, ms-ESM struggles to extract structural information from unzipped residues, making the training task too simplistic and hindering the acquisition of meaningful knowledge.
>   - Results indicate that, while both atom type and structural information are vital for atomic-scale details, structural information holds greater importance.
> - Ablation Study on Pre-training Data:
>   - A significant decrease in model performance is observed when either molecular data or protein data is excluded.
>   - Interestingly, the removal of protein data leads to a more substantial performance decline than omitting molecular data.
>     - Insights:
>       - When the model is not trained with protein data, it quickly forgets protein-related knowledge, resulting in a notable decline in overall performance.
>       - The model can still learn some atomic scale information through unzip operations without molecular data. This explains why the model performs better without molecular data compared to the scenario without protein data.
>       - This experiment effectively demonstrates the beneficial impact of the unzip operation in modeling atomic-scale information.
> - Ablation Study on Unzip Operation: As analyzed in Q1, the model performance drops significantly after removing the unzip operation, which illustrates its importance for modeling unified semantics.
>
> Table D. Ablation study
> |  | MSE ↓ | R^2 ↑ |
> | --- | --- | --- |
> | Vanilla ms-ESM | 0.627 | 0.546 |
> | w/o ASPE | 0.639(+0.012) | 0.537(-0.009) |
> | w/o RSPE in atoms  | 0.627(+0.0) | 0.547(+0.001) |
> | w/o masked atom type loss | 0.642(+0.015) | 0.535(-0.011) |
> | w/o pair-wise distance recovery loss | 0.645 (+0.018) | 0.533(-0.013) |
> | w/o molecular data | 0.648 (+0.021) | 0.531(-0.015) |
> | w/o protein data | 0.708(+0.081) | 0.487(-0.059) |
> | w/o unzip operation | 0.638 (+0.011) | 0.538(-0.008) |
>
> [1] Wang S, Guo Y, Wang Y, et al. Smiles-bert: large scale unsupervised pre-training for molecular property prediction. ACM-BCB 2019
>
> [2] Honda S, Shi S, Ueda H R. Smiles transformer: Pre-trained molecular fingerprint for low data drug discovery. arXiv 2019
>
> [3] Chithrananda S, Grand G, Ramsundar B. ChemBERTa: large-scale self-supervised pretraining for molecular property prediction. arXiv 2020
>
> [4] Zhang X C, Wu C K, Yang Z J, et al. MG-BERT: leveraging unsupervised atomic representation learning for molecular property prediction. Briefings in bioinformatics 2021
>
> [5] Xue D, Zhang H, Xiao D, et al. X-MOL: large-scale pre-training for molecular understanding and diverse molecular analysis. bioRxiv 2020
>
> [6] Guo Z, Sharma P, Martinez A, et al. Multilingual Molecular Representation Learning via Contrastive Pre-training. ACL 2022
>
> [7] Rong Y, Bian Y, Xu T, et al. Self-supervised graph transformer on large-scale molecular data. NeurIPS 2020
>
> [8] Li P, Wang J, Qiao Y, et al. Learn molecular representations from large-scale unlabeled molecules for drug discovery. arXiv 2020
>
> [9] Zhang Z, Liu Q, Wang H, et al. Motif-based graph self-supervised learning for molecular property prediction. NeurIPS 2021
>
> [10] Li P, Wang J, Qiao Y, et al. An effective self-supervised framework for learning expressive molecular global representations to drug discovery. Briefings in Bioinformatics 2021
>
> [11] Wang Y, Magar R, Liang C, et al. Improving molecular contrastive learning via faulty negative mitigation and decomposed fragment contrast. Journal of Chemical Information and Modeling 2022
>
> [12] Fang Y, Zhang Q, Yang H, et al. Molecular contrastive learning with chemical element knowledge graph. AAAI 2022
>
> [13] Lin X, Xu C, Xiong Z, et al. PanGu Drug Model: learn a molecule like a human. bioRxiv 2022

---

> ### Author Response · Authors · 2023-11-18
> **Response to Reviewer 54HD (part 5)**
>
> [14] Liu S, Wang H, Liu W, et al. Pre-training molecular graph representation with 3d geometry. ICLR 2022
>
> [15] Li S, Zhou J, Xu T, et al. Geomgcl: Geometric graph contrastive learning for molecular property prediction. AAAI 2022
>
> [16] Zhu J, Xia Y, Wu L, et al. Unified 2d and 3d pre-training of molecular representations. KDD 2022
>
> [17] Stärk H, Beaini D, Corso G, et al. 3d infomax improves gnns for molecular property prediction. ICML 2022
>
> [18] Fang X, Liu L, Lei J, et al. Geometry-enhanced molecular representation learning for property prediction. Nature Machine Intelligence 2022
>
> [19] Zhou G, Gao Z, Ding Q, et al. Uni-Mol: a universal 3D molecular representation learning framework. ICLR 2023.
>
> [20] Luo S, Chen T, Xu Y, et al. One transformer can understand both 2d & 3d molecular data. ICLR 2023
>
> [21] Zaidi S, Schaarschmidt M, Martens J, et al. Pre-training via denoising for molecular property prediction. arXiv 2022
>
> [22] Liu S, Guo H, Tang J. Molecular geometry pretraining with se (3)-invariant denoising distance matching. ICLR 2023
>
> [23] Jiao R, Han J, Huang W, et al. Energy-motivated equivariant pretraining for 3d molecular graphs. AAAI 2023
>
> [24] Lample, Guillaume, and Alexis Conneau. Cross-lingual language model pretraining. arXiv 2019
>
> [25] Krishna, Rohith, et al. Generalized Biomolecular Modeling and Design with RoseTTAFold All-Atom. bioRxiv 2023

---

> ### Author Response · Authors · 2023-11-21
> **Response to Reviewer 54HD (part 6)**
>
> Dear Reviewer 54HD, we are deeply appreciative of your insightful comments, which have undoubtedly enhanced the quality of our manuscript.
>
> Please let us know if our response and the revised manuscript have successfully resolved your concerns. We welcome any further questions or suggestions you may have.
>
> Thank you for your valuable time and consideration.

---

> > ### Comment · Reviewer_54HD · 2023-11-21
> >
> > I’d like to thank the authors’ efforts in addressing my questions during rebuttal. However, my questions about the soundness, novelty, experimental evaluation of the paper remain. Here is the detailed response:
> > >**Q1: Design of the unzipped operation.**
> > 1. The authors claim that structural information can be learned through pairwise distance recovery on unzipped residues. However, the pairwise distances between atoms (with noise on coordinates) are provided in atom-scale position encoding. This means that the task is essentially learning to denoise the pairwise distance, which focuses on fine-grained features instead of global positional information about atoms. Also, the authors should provide more justification behind this design: Why corrupting coordinates instead of distances? Why denoising on distances instead of coordinates? Can we directly predict the coordinate for each atom or the side-chain angles for each residue?
> > 2. I don’t think that the model can learn associations between proteins and molecules during pre-training, as it only mixes protein-only and molecule-only data during pre-training and does not include any protein-molecule interaction data.
> > 3. In my opinion, the ablation study shown in Table D is not convincing. In the table, the only operation with significant impact is “training w/o protein data”.
> > 4. I think it is not proper to use RosettaFoldAA as a reference to support the design of the unzipped operation. First, RFAA studied a very different problem and **they did use multiple kinds of protein-molecule data for pre-training, which was not done by this paper**. Second, even in their paper, they did not include an ablation study for the “atomization” operation. Please correct me if I’m wrong.
> >
> > >**Q2: Novelty of the multi-scale position encoding.**
> >
> > I still don’t think the authors do a good job on designing multi-scale position encoding. The design is trivial in concept. Besides, as shown in Table 7, the “carefully designed” residue-scale position encoding does not have large effects on the performance.
> >
> > >**Q3,4: Experimental results**
> >
> > As the paper is titled “for universal molecular modeling”, I expected that the model was able to achieve improvements on protein-only, molecule-only and protein-molecule tasks. That’s also why I expected mixing protein-only and molecule-only data would be beneficial for each other.
> >
> > However, during rebuttal, the author claims their focus is just on protein-molecule tasks, which is contradictory to the presentation in the paper. I would suggest the authors to rethink about the contribution of the paper and put more stress on how the designed pre-training algorithm can help protein-molecule tasks. Also, if the focus is on protein-molecule tasks, more downstream tasks should be considered, e.g., protein-ligand binding in Atom3D.
> >
> > >**Q6: Structure input during fine-tuning**
> >
> > The authors claim “structure information will not be used for protein-only tasks“. This means that the model does not use atom-scale position encoding in downstream tasks. If so, please explicitly describe more details of these discrepancy between pre-training and fine-tuning in the main paper.
> >
> > >**Q8: Potential data leakage.**
> >
> > First, to remove the potential data leakage, you cannot only remove the overlap between pre-training and downstream datasets. Instead, the common practice should be remove all the sequences with high sequence similarity with the pre-training datasets.
> > Second, the real problem behind the data leakage is that the pre-training objective includes pairwise distance recovery. This means that the proposed methods use more data for training on the contact prediction than other baselines, which leads to unfair comparison.

---

> ### Comment · Reviewer_54HD · 2023-11-21
>
> >**General comments**
>
> I find it difficult to justify the experimental contribution of the paper (reasons in my reply to Q3,4). I think the paper shows convincing experimental results only if (1) the idea of universal pre-training can consistently improve any protein language models or (2) with the proposed pre-training methods, we can achieve the state-of-the-art performance on the problems that the paper want to study.
>
> For (1), I would expect more experiments on different language models to show the consistent improvement, and larger performance improve over the baselines.
> For (2), I would expect more baselines included in the benchmark on the studied problems. In Tables 1&3, there are only 3-4 baselines considered. One possible reason is that the domain is still underexplored and some baselines are difficult to reproduce. In this case, I suggest the authors to implement some straightforward baselines, e.g., different protein language models, different molecule encoders and fusion mechanisms.
>
> Moreover, I suggest the authors to carefully use the word “state-of-the-art”. Clearly, ESM-2-35M is not a state-of-the-art method on contact prediction. But the authors claim “ms-ESM is on par with
> the state-of-the-art in protein-only tasks” in the protein-only tasks.
>
>
> Overall, I’d like to express my gratitude again in the authors’ long and detailed response. I believe there are still large space in the paper to improve. Therefore, I keep my score unchanged.

---

> ### Author Response · Authors · 2023-11-21
> **Reply to Reviewer 54HD (Part 1)**
>
> Thank you very much for the your patient response. I would also like to express my gratitude once again for the numerous valuable suggestions you provided for this work. Regarding your questions, our response is as follows:
>
> > **I don’t think that the model can learn associations between proteins and molecules during pre-training, as it only mixes protein-only and molecule-only data during pre-training and does not include any protein-molecule interaction data.**
>
> * We do not consider it a drawback that we do not use protein-molecule interaction data. On the contrary, we believe that designing a mechanism for the model to learn the protein-molecule correlation on mixed protein-only and molecule-only data is a more meaningful approach. In many cases, collecting protein-molecule interaction data is challenging, and the data volume is significantly smaller than that of protein-only and molecule-only data. Therefore, exploring how to enhance the model's performance on protein-molecule tasks without using protein-molecule interaction data is a meaningful topic.
> * The unzip operation plays a crucial role in providing the model with atomic-scale information about proteins. Considering that most small molecule data is represented at the atomic scale, the unzip operation allows the model to uniformly represent both types of data at the atomic scale. This, in turn, assists the model in better capturing the correlation between the two. We have experimentally validated the significance of using the unzip operation to achieve this objective.
> * Our model is not in conflict with protein-molecule interaction data. In fact, to further enhance the model's performance in protein-molecule tasks, we can also fine-tune the model with protein-molecule interaction data to achieve additional performance improvements. However, this would create an unfair comparison between the model and the ESM-2+Uni-Mol combination. What we aim to reveal is the performance advantage of this unified modeling approach under similar training conditions (similar training data, similar training tasks) to ESM-2+Uni-Mol.
>
> > **Why corrupting coordinates instead of distances?**
>
> In fact, you've raised a very good question. The reason we chose to add noise to coordinates rather than distances is primarily based on the following two considerations:
> * For structural information, adding noise to coordinates often has a similar effect as adding noise to distances. In both cases, the goal is to disrupt a structural information and require the model to reconstruct it. From this perspective, adding noise to coordinates can achieve this objective. Therefore, we opted for the approach of adding noise to coordinates.
> * In the classic molecular 3D structure representation learning work, Uni-Mol[19], the authors also employed the method of adding noise to coordinates and explored the optimal selection of noise parameters. Moreover, Uni-Mol has demonstrated the effectiveness of this approach. Considering that we are also engaged in pre-training for molecular structure representation learning, we referenced their approach.
>
> >**Can we directly predict the coordinate for each atom or the side-chain angles for each residue?**
>
> We have also attempted to adopt the approach of coordinate prediction used in works like Uni-Mol. However, in practice, we encountered the following two issues:
> * Attempting to make the model predict coordinates necessitates dealing with SE(3) equivariance issues, requiring the introduction of a more complex equivariant head to predict atomic coordinates. This would make the model structure more intricate, and the added complexity did not significantly improve the model's performance. Consequently, we decided to remove this loss.
> * After adding this training objective, the training process became more unstable, making it more prone to issues such as gradient explosions. Therefore, to enhance training stability, we also chose not to retain this loss.
>
> >**The real problem behind the data leakage is that the pre-training objective includes pairwise distance recovery.**
>
> One clarification that needs to be made is that our pairwise distance recovery is limited to atoms within the same residue. We do not provide the model with structural information for atoms across different residues, nor do we require the model to predict atomic distances across residues. Contact prediction, on the other hand, is more focused on distances between atoms in different residues. Therefore, training the model on the structural information within an individual residue will not lead to additional advantages for contact prediction tasks.

---

> ### Author Response · Authors · 2023-11-21
> **Reply to Reviewer 54HD (Part 2)**
>
> >**The “carefully designed” residue-scale position encoding does not have large effects on the performance**
>
> In Table 7, we specifically removed the residue-scale position encoding for the atoms while retaining the residue-scale position encoding for each residue. Therefore, the result under the "w/o RSPE in atoms" category does not imply that residue-scale position encoding is ineffective. In fact, residue-scale position encoding is crucial for the model to distinguish between different residues, even though we removed it from the individual atoms. The role of residue-scale position encoding is similar to sequence position embeddings in Transformer. Thus, without residue-scale position encoding, the model would be unable to discern the positions of different residues, significantly impacting its performance.
>
> >**This means that the model does not use atom-scale position encoding in downstream tasks.**
>
> Firstly, it's important to clarify a misunderstanding. Atom-scale position encoding is indeed utilized in downstream tasks. For molecular data, atom-scale position encoding is a crucial factor in distinguishing between different atomic positions. Therefore, we provide atom-scale position encoding of molecules for the model in all downstream tasks related to molecular data. The exception lies in our protein-only tasks, where we do not provide structural information for protein. The construction of atom-scale position encoding requires structural information, which is why we did not use atom-scale position encoding in all protein-only tasks. This is the specific reason for not using atom-scale position encoding only in protein-only tasks. Apologies for the misunderstanding. We will provide additional clarification in the paper to eliminate any misconceptions.
>
>
> Thank you once again for the numerous valuable suggestions you provided, especially regarding the selection of experimental tasks. In the future, we will explore more protein-molecule tasks to offer a more comprehensive evaluation of the model's performance. Additionally, we apologize again for any misunderstandings caused. We will modify the relevant statements in the paper to be more accurate and reduce the potential for confusion.

---

> > ### Comment · Reviewer_54HD · 2023-11-22
> >
> > I'd like to thank the authors' detailed response, which clarifies some misunderstanding in the paper. I believe most of these are due to the lack of details in the paper. Also, my questions about the experimental contribution of the paper remain. So I hold my opinition that there are still large space in the paper to improve. Therefore, I keep my score unchanged.

---

> > > ### Author Response · Authors · 2023-11-22
> > >
> > > Thank you for reading our response and providing feedback. If you have any further questions, please feel free to continue the conversation.

---

### Official Review · Reviewer_p5PD · 2023-10-31

**Soundness:** 3 good
**Presentation:** 3 good
**Contribution:** 2 fair
**Rating:** 6
**Confidence:** 3

**Summary:**

In the context of protein language models, this paper attempts to fuse the information contained at the residue scale with the one contained at the atom scale (i.e. the structure of the resides themselves) to produce better models. The protein sequence is thus represented by inserting the set of atoms constituting a certain residue inbetween the residues. To accomodate this change, the authors propose ad hoc position encodings depending on the scale (atom or residue). The method is tested in several (protein-molecule, protein-only, and molecule-only) tasks with good overall performances.

**Strengths:**

- the proposed idea makes sense intuitively, and appears to be effective across different tasks.

- the experiments are well thought and the results seem convincing, both in depth and width.

**Weaknesses:**

- the technical novelty is limited; I understand code-switching is not novel but I value that it is ported to this field. However, the rest of the techniques used in this paper (like RoPE, or the transformer architecture) are not novel.

- Lack of detail on certain topics. For example, in Section 2.2, the ORDER procedure is not explained (there is a referral to Appendix A, where however I didn't find an explanation). Similarly, the "Atom Scale Position Encoding" section is not informative with lots of unintroduced symbols.

- The "slight modification of the Transformer" in section 2.4 appears poorly justified or at least needs more clarification. Why is $E^A$ added to the standard attention? What happens if it's not added?

- Also in Section 2.4, my guess is that the scaling by $\sqrt{d_k}$ gets disrupted by the $E^A$ term. Can you comment on this latter point?

- The ablation study in Table 7 shows almost no improvement from vanilla ESM to the "w/o RSPE in atoms" variant.

**Questions:**

Mostly related to the weaknesses, see above. On the more discussive side:
- did you consider the idea of representing the structure of the residue with a graph neural network? What could be the up/downsides of this approach?
- it is unclear how the sequence length is affected by the addition of the atoms constituting the residues. Can you detail how long are the sequences you deal with, and how much their lengths increase by adding the atoms?

---

> ### Author Response · Authors · 2023-11-18
> **Response to Reviewer p5PD (part 1)**
>
> Thanks for your constructive comments and suggestions. Following the comments, we clarify the details of our method and address these issues as follows. Further comments are welcome!
>
> **Q1**: Technical novelty is limited.
>
> **A1**: Please refer to the Q2 of the general response for more discussion. In brief, our work is novel for three reasons: i) we take the construction of a model (not exisiting techniques) which can process proteins and molecules uniformly as the core contribution of our work. ii) Unified molecular modeling topic is the cutting edge of AI biological application emerging recently. iii) Adopting existing techniques to our multi-scale unified molecular modeling is non-trivial
>
> **Q2**: Lack of detail on certain topics, e.g., ORDER procedure and unintroduced symbols in Atom Scale Position Encoding.
>
> **A2**: Thank you for bringing this to our attention, which enhances the clarity of our paper. While we briefly touched upon these details in our manuscript (Appendix A and Sections 2.2, 2.3), we will now introduce them more comprehensively in the following paragraphs and **incorporate them into the revised paper.**
> - Regarding the ORDER procedure, as indicated in Appendix A (the final line of the Hyperparameters section), we adopt the default order from PDB (protein) and SDF (molecule) files for assigning order to the atom set.
>   - To elaborate, PDB and SDF serve as standard formats for describing atomic structures of proteins and small molecules, respectively. In both formats, atoms follow specific sorting principles. In our study, we directly utilize the sorted atoms for ease of implementation.
>   - It is important to note that, given our atom-scale position encoding employs Euclidean distance to describe positional relationships, the permutation of atom order does not impact our pre-training model.
>
> - For atom scale position encoding, following Uni-Mol [1], we use Euclidean distance and Gaussian kernel to encode the 3D position of atoms. For $h_i, h_j \in \bar{X}$ (both $h_i$ and $h_j$ can be a residue or atom in code-switch protein sequence $\bar{X}$, Section 2.2), their atom scale position encoding $e_{ij}^{A}$ can be calculate as:
> $$
> e_{ij}^{A} = \begin{cases}
> 0 & h_i \in \bar{R} ~or ~h_j \in \bar{R} \\\\
> Gaussian(Dis(c_i, c_j)) & otherwise, c_i = Coord(h_i),c_j = Coord(h_j) \end{cases}
> $$
>   where $\bar{R}$ is the set of all the residues in $\bar{X}$ (Section 2.2), $Coord$ extracts coordinates from atoms (Section 2.2), $Dis$ calculates the Euclidean distance between two coordinates (Section 2.2), and $Gaussian$ is the Gaussian kernel (Section 2.3).
>
> **Q3**: Why is $E^A$ added to the standard attention? What happens if it's not added?
>
> **A3**:
> - To enhance the awareness of the 3D position relationship between atoms, we incorporate $E^A$ into the attention layer of the vanilla Transformer, inspired by the approach in Uni-Mol [1], a widely recognized technique in geometric molecular modeling [2, 3, 4] known for its efficacy in various downstream tasks. Specifically, when two atoms are in close proximity in 3D space, $E^A$ boosts the attention value between them, subsequently encoding this positional relationship into the Transformer.
> - Ablation studies detailed in our paper (Section 3.4) offer empirical evidence affirming the effectiveness of this 3D position encoding. In the absence of $E^A$, ms-ESM struggles to differentiate molecules with identical atoms but distinct structures, e.g., isomer [5], resulting in a performance drop from 0.546 to 0.537 (refer to Table 7 in our paper).
>
> **Q4**: Also in Section 2.4, my guess is that the scaling by $\sqrt{d_k}$ gets disrupted by the $E^A$ term. Can you comment on this latter point?
>
> **A4**:
> - That's an excellent question! Currently, we haven't encountered any issues with disrupted scaling. However, we acknowledge the importance of this matter and plan to delve deeper into it in our future research.
> - It's worth noting that the inclusion of encoding (as represented by $E^A$ in our paper) into attention mechanisms is a widely adopted practice across various models in different domains. For instance, both T5 [9] and Swin Transformer [10] utilize this operation in language and vision modeling, respectively. Similarly, in the realm of molecule modeling, Transformer-M [2] and Uni-Mol [1] also incorporate this operation, demonstrating excellent performance in their respective domains.

---

> ### Author Response · Authors · 2023-11-18
> **Response to Reviewer p5PD (part 2)**
>
> **Q5**: Table 7 shows almost no improvement from vanilla ESM to the "w/o RSPE in atoms" variant.
>
> **A5**: Thank you for your insightful question. **The model's behavior in this ablation experiment aligns with our expectations, and residue-scale positional encoding in atoms aids the model in distinguishing atoms from different residues.** Here's an analysis of the phenomenon:
> - The model's behavior aligns with our expectations:
>   - In protein-molecule tasks, the model only requires protein sequences at the residue scale as input. Consequently, the addition of residue-scale positional encoding to atoms has no impact on the model's performance.
> - Despite the lack of performance improvement from adding residue-scale positional encoding to atoms, we choose to retain this feature in the model. This decision is driven by the understanding that residue-scale positional encoding aids the model in distinguishing atoms from different residues. In the future, we will also explore more diverse and effective approaches to integrate residue-scale information into atomic representations.
>
> **Q6**: Did you consider the idea of representing the structure of the residue with a graph neural network? What could be the up/downsides of this approach?
>
> **A6**:
> - Indeed, we did consider the idea of employing a graph neural network (GNN) for representing residue structures. However, in alignment with previous research, we opted for a Transformer-based architecture in this paper. While we acknowledge the promise of using GNN for residue representation, we chose the Transformer due to its established success in molecular modeling.
> - The Transformer-based architecture is a well-established and reliable choice in molecular modeling, serving as the foundation for many successful methods such as the ESM series and Uni-Mol. Utilizing this architecture, aligned with these well-established methods, offers a more direct and secure approach. It facilitates the incorporation of their experiences into our work, mitigating the risks associated with project development. Choosing a GNN as the backbone might introduce unnecessary complexities, such as the need to train the entire model from scratch.
> - We recognize the strengths of GNN in capturing detailed structural information. For instance, GNN allows us to define edge features to capture chemical bonds in residue structures. Furthermore, it enables us to specify aggregation strategies for modeling information related to functional groups and motifs. Despite not being the focus in this paper, we believe GNN holds promise for future exploration in representing residue structures.
>
> **Q7**: Can you detail how long the sequences you deal with, and how much their lengths increase by adding the atoms?
>
> **A7**:
> - As each residue in the AlphaFold dataset typically consists of 7.9 atoms, we choose a 6.5% unzip ratio (refer to Appendix A, Table 8). Consequently, the length of each protein sequence is expected to increase by an average of 7.9 * 6.5%, resulting in a 51.35% increase. This implies that the length of each protein sequence after unfolding will be approximately 1.51 times the original.
> - In light of this length increase, we make an adjustment to the maximum sequence length permissible for the model, transitioning from ESM-2's 1024 to 2048.
>   - In instances where the sequence length exceeds the model's updated maximum after unzipping, a restriction is enforced. One unzipped residue is randomly chosen and the unzipping process is halted until the length aligns with the model's predefined limit.
>
> [1] Zhou G, Gao Z, Ding Q, et al. Uni-Mol: a universal 3D molecular representation learning framework. ICLR 2023
>
> [2] Luo S, Chen T, Xu Y, et al. One transformer can understand both 2d & 3d molecular data. ICLR 2023
>
> [3] Yu Q, Zhang Y, Ni Y, et al. Unified Molecular Modeling via Modality Blending. arXiv 2023
>
> [4] Raffel C, Shazeer N, Roberts A, et al. Exploring the limits of transfer learning with a unified text-to-text transformer. The Journal of Machine Learning Research 2020
>
> [5] Wikipedia. Isomer. https://en.wikipedia.org/wiki/Isomer. Wikipedia 2023
>
> [6] Lin Z, Akin H, Rao R, et al. Evolutionary-scale prediction of atomic-level protein structure with a language model. Science 2023
>
> [7] Nijkamp E, Ruffolo J A, Weinstein E N, et al. ProGen2: exploring the boundaries of protein language models. Cell Systems 2022
>
> [8] Madani A, Krause B, Greene E R, et al. Large language models generate functional protein sequences across diverse families. Nature Biotechnology 2023
>
> [9] Raffel, Colin, et al. Exploring the limits of transfer learning with a unified text-to-text transformer. JMLR 2020.
>
> [10] Liu, Ze, et al. Swin transformer: Hierarchical vision transformer using shifted windows. CVPR 2021.

---

> ### Author Response · Authors · 2023-11-21
> **Response to Reviewer p5PD (part 3)**
>
> Dear Reviewer p5PD, we sincerely appreciate your thoughtful feedback, which has significantly improved our manuscript.
>
> We have carefully considered your concerns and believe that our response and the updated manuscript adequately address them. Nevertheless, we are open to further discussions and revisions if you have any additional feedback or suggestions.
>
> Thank you for your time and dedication to reviewing our work.

---

> ### Author Response · Authors · 2023-11-22
> **Seeking Feedback on Our Response**
>
> Thank you for your review. We've addressed the concerns raised in our response. Could you please verify if our explanations meet your expectations? We're open to further discussion and welcome any questions you may have.

---

> > ### Comment · Reviewer_p5PD · 2023-11-22
> > **Thanks**
> >
> > Thank you for your effort. While my concerns are mostly addressed, having read all the other reviewer's concerns, I decided to leave my score unchanged.

---

> > > ### Author Response · Authors · 2023-11-22
> > >
> > > Thank you for reading our response and providing feedback. If you have any further questions, please feel free to continue the conversation.

---

### Official Review · Reviewer_VsVJ · 2023-11-02

**Soundness:** 3 good
**Presentation:** 3 good
**Contribution:** 2 fair
**Rating:** 6
**Confidence:** 3

**Summary:**

This paper proposes a multi-scale language model for protein and small molecule modeling. The author combines masked language modeling and pair-wise distance recovery to pretrain the model. The authors present competitive results against baselines on multiple tasks.

**Strengths:**

1. The paper is clearly written.
2. The multiscale modeling technique is novel.

**Weaknesses:**

1. The proposed method could not outperform baselines, as shown in Table.5 and Table.6.
2. Insufficient experiments regarding molecular representation learning affects the significance of the paper.

**Questions:**

1. Could you provide head-to-head comparisons against Unimol on molecular representation learning tasks, such as BBBP,  BACE, Tox21?
2. How about scaling the 35 million model to 650 million? Could you provide the corresponding results?

---

> ### Author Response · Authors · 2023-11-18
> **Response to Reviewer VsVJ (part 1)**
>
> Thank you very much for your good suggestions. Following the comments, we clarify the details of our method and address these issues as follows. Further comments are welcome!
>
> **Q1**: The proposed method could not outperform baselines, as shown in Paper Table.5 and Paper Table.6.
>
> **A1**: Please refer to the Q1 of the general response for more discussion. Briefly, i) as already mentioned in the answer to Q1 in the general response, this paper focuses on unified multi-scale modeling, aiming to obtain better results on protein-molecule tasks  instead of improving molecule/protein tasks universally. ii) We report molecule-only and protein-only results to verify that ms-ESM does not suffer from significant performance drops on separated molecular and protein modeling tasks.
>
> **Q2**: Insufficient experiments regarding molecular representation learning affects the significance of the paper. Could you provide head-to-head comparisons against Unimol on molecular representation learning tasks, such as BBBP, BACE, Tox21?
>
> **A2**: Thank you very much for your suggestions. We did a quick run to fix your concern. We don't include the results you mentioned because this paper focuses on unified multi-scale modeling for obtaining better results on protein-molecule tasks, and we only report results on  several selected representative molecule-only tasks to verify that the proposed ms-ESM does not suffer from significant performance drops on separated molecular modeling tasks. However more experimental results on molecular representation will be definitely helpful for the analysis. The added results are shown in Table A and we will also add them to the manuscript.
>
> Table A: Molecule-only tasks
> |  | QM7 ↓ | QM8 ↓ | QM9 ↓ | BACE↑ | BBBP↑ | TOX21↑ | PCBA↑ | SIDER↑ | HIV↑ | MUV ↑ |
> | --- | --- | --- | --- | --- | --- | --- | --- | --- | --- | --- |
> | Uni-Mol w/o H | 58.8678 | **0.0160** | **0.00540** | 83.20 | **71.52** | **78.92** | **88.12** | 57.71 | **78.3** | 72.0 |
> | ms-ESM | 60.8755 | 0.0166 | 0.00590 | **83.52** | 67.41 | 75.39 | 86.15 | **63.59** | 74.9 | **72.6** |
> | ms-ESM 8M | **52.8018** | 0.0173 | 0.00630 | 75.97 | 67.30 | 76.33 | 85.90 | 62.64 | 74.3 | 70.73 |
>
> **Q3**: How about scaling the 35 million model to 650 million? Could you provide the corresponding results?
>
> **A3**: We recognize the importance of understanding how changes in model size affect its performance. However, due to the substantial computational resources required to train the 650M model—an estimated 16 A100 GPUs over two weeks—we opted to train a smaller, 8M model to swiftly gather results and examine the impact of model scaling. Our intention is to extend this initial research by training larger models and including those findings in the final version of the paper.
>
> Our observations are as follows:
> - For tasks involving both proteins and molecules (as shown in Table B) and for tasks involving only proteins (as shown in Tables C and D), we found a clear trend: as the model size grows, there is a noticeable enhancement in performance.
> - For tasks involving only molecules (as presented in Table A): 1) The larger model tends to perform better on the majority of these tasks, correlating increased model size with improved outcomes. 2) However, a few tasks saw superior performance from the smaller model, notably QM7 and TOX21, suggesting that the larger model may be susceptible to overfitting in some cases.
>
> In conclusion, **we generally observed that model performance tends to improve with an increase in the number of parameters**, with a few exceptions. We plan to validate these findings by evaluating the 650M parameter model upon its completion to further test the performance of a larger-scale model.
>
> Table B: The new results on enzyme-substrate affinity regression task
> | Method | Protein Pre-training | Molecule Pre-training | MSE↓ | R^2↑ | Pearson↑ |
> | --- | --- | --- | --- | --- | --- |
> | XGBoost | ESM-2 35M | Uni-Mol 48M | 0.652 | 0.528 | 0.727 |
> | ProSmith | ESM-2 35M | Uni-Mol 48M | 0.642 | 0.536 | 0.733 |
> | XGBoost | ms-ESM 35M | ms-ESM 35M | 0.623 | 0.548 | 0.742 |
> | ProSmith | ms-ESM 35M | ms-ESM 35M | **0.599** | **0.566** | **0.753** |
> | XGBoost | ms-ESM 8M | ms-ESM 8M | 0.63188 | 0.54213 | 0.7269 |
> | ProSmith | ms-ESM 8M | ms-ESM 8M | 0.61761  | 0.55247  | 0.7344 |
>
> Table C: New experimental results on contact map prediction. SR: Short Range, MR: Medium Range, LR: Long Range.
> |   |  SR P@L|  SR P@L/2 | SR P@L/5 | MR P@L | MR P@L/2 |  MR P@L/5 | LR P@L | LR P@L/2 | LR P@L/5 |
> | --- | --- | --- | --- | --- | --- | --- | --- | --- | --- |
> | ESM2 35M | 0.197 | 0.293 | 0.458 | 0.223 | 0.318 | 0.452 | **0.297** | **0.385** | **0.492** |
> | msESM 35M | **0.206** | **0.311** | **0.481**  | **0.225** | **0.320** | **0.453**  | 0.289 | 0.375  | 0.479 |
> | msESM 8M | 0.153 | 0.213 | 0.310 | 0.143 | 0.191 | 0.259 | 0.153 | 0.197 | 0.256 |

---

> ### Author Response · Authors · 2023-11-18
> **Response to Reviewer VsVJ (part 2)**
>
> Table D: New experimental results on secondary structure prediction.
> |  | SS3 cb513 |SS3 ts115 |SS3 casp12 |SS8 cb513 |SS8 ts115 |SS8 casp12 |
> | --- | --- | --- | --- | --- | --- | --- |
> | ESM2 35M | **0.8043** | **0.8248** | **0.7403** | **0.6491** | **0.7013** | **0.6101** |
> | msESM 35M | 0.7923 | 0.8136 | 0.7381 | 0.6326 | 0.6856 | 0.6040 |
> | ms-ESM 8M | 0.7521 | 0.7888 | 0.7265 | 0.5948 | 0.6546 | 0.5841 |

---

> ### Author Response · Authors · 2023-11-21
> **Response to Reviewer VsVJ (part 3)**
>
> Dear Reviewer VsVJ, we are grateful for your insightful feedback, which has significantly improved our manuscript.
>
> Please let us know if we have adequately addressed your concerns. If not, we are open to further questions and revisions.
>
> Thank you for your time and consideration.

---

> ### Author Response · Authors · 2023-11-22
> **Seeking Feedback on Our Response**
>
> Appreciate your review! We've addressed the raised concerns in our response. Could you confirm if it aligns with your expectations? Open to discussing any lingering questions or points you'd like to explore further.

---

> ### Comment · Reviewer_VsVJ · 2023-11-22
> **Response to authors**
>
> Upon first reading of this paper, I greatly appreciate its innovation, and I am inclined to give an acceptance score. After reading it carefully, I found there is still large room for improvement, although the authors have made great efforts to address the issues.
>
> - The scaling up of the model has not been addressed, and it is currently unclear how the proposed method performs at different model scales.
>
> - The performance on protein-only and molecule-only tasks is currently poor. Fourtunately, the authors have found excellent performance on protein-molecule complex tasks. However, there is a lack of crucial ablation experiments on protein-molecule complex tasks: ESM2 (protein) + msESM (molecule) and msESM (protein) + UniMol (molecule).
>
> - Have you attempted to randomly modify the order of atoms during the training/testing phase? Does it have any impact on the experimental results?
>
> - There are also many other questions raised by the other reviewers.
>
> I will raise the score to 6 to encourage authors' efforts, but it does not mean that I am satisfied with this work at this stage. If the authors intend to publish a paper, modifications to the experiments are needed to highlight the strengths and provide further analysis for the protein-molecule complex task. If the authors aim to satisfy my expectations, more exploration is required. Nonetheless, I appreciate your innovative ideas and efforts. If the authors truly believe that the proposed method has potential, it may require more time to fully immerse themselves in its development until it gains widespread recognition. After all, this is a pre-training work, and the workload involved is undoubtedly massive.

---

> > ### Author Response · Authors · 2023-11-23
> > **Response to Reviewer VsVJ**
> >
> > Thank you very much for patiently reading our paper and providing feedback. We also appreciate your positive comments and valuable suggestions for this paper. Regarding the questions you raised in your response, here are our replies:
> >
> > >**The scaling up of the model has not been addressed, and it is currently unclear how the proposed method performs at different model scales.**
> >
> > - Model scaling is indeed a core concern for us, and we are currently attempting to train a 650M model. However, due to limitations in computational resources, we have not yet completed the entire training process. In the final version of the paper, we will report the performance of larger models. Based on the interim training results, it appears that scaling this model is beneficial for improving model performance, especially for protein-only tasks, where the performance gains from model scaling are more pronounced. For molecular data, the performance gains from model scaling are relatively smaller. Regardless of the final results, we will report these findings in the final version of the paper. Thank you for your valuable suggestions.
> >
> > >**there is a lack of crucial ablation experiments on protein-molecule complex tasks: ESM2 (protein) + msESM (molecule) and msESM (protein) + UniMol (molecule)**
> >
> > - Due to time constraints, we are unable to promptly report the results of these ablation experiments. However, we will include the results of these ablation experiments in the final version of the paper. Based on our experience, the performance improvement in the joint protein-molecule task comes primarily from the unified modeling of the semantic space rather than an enhancement in the modeling capability of a single model on a specific type of data. The impact of unified semantic modeling can also be demonstrated from the results in Section 3.5 of the paper, showing that the representations learned by a single unified model for different types of data are more consistent than those learned by multiple individual models. Therefore, we believe that the combination of msESM (protein) + msESM (molecule) has an advantage in tasks of this nature.
> >
> > >**Have you attempted to randomly modify the order of atoms during the training/testing phase? Does it have any impact on the experimental results?**
> >
> > - This is a great question. In fact, since we did not provide "sequential information" for the atoms at the atomic scale but only their spatial positions, shuffling the order of these atoms will not significantly affects the model's performance. In the attention mechanism, the most crucial way to distinguish between different input tokens is through position embedding. When shuffling the order of atoms, we do not change the position embedding corresponding to different atoms. Therefore, shuffling the order of atoms has a relatively small impact on the model's performance.
> >
> > We hope these discussions help address some of your questions. We will promptly complete any additional experimental results that need to be supplemented and report these results and further analysis in the final version of the paper. Once again, we appreciate your affirmation and suggestions, and if you have any further questions, please feel free to discuss them.

---

### Official Review · Reviewer_ZacV · 2023-11-02

**Soundness:** 2 fair
**Presentation:** 2 fair
**Contribution:** 2 fair
**Rating:** 5
**Confidence:** 4

**Summary:**

This work aims to appy the powerful protein language models (ESM) to the applications of both small molecules and proteins.


Specifically, the authors provides a multi-scale ESM (ms-ESM) model for the unified molecular modeling. The ms-ESM model can take both protein sequences and molecules with 3D coordinates as input.


The model is pretrained using both protein dataset (AlphaFoldDB) and molecule dataset (from uni-mol). Each residue in a protein can also be unzipped to several atoms. The pre-training tasks are masked language modeling (MLM) and pair-wise distance recovery.


The architecture of ms-ESM is very similar to ESM, and one main difference is that the atom scale position encoding (Euclidean distance + Gaussian kernel) is used as a bias term in the attention layers.


The proposed ms-ESM is evaluated on protein-molecule tasks, protein-only tasks, and molecule-only tasks.

**Strengths:**

The idea seems interesting. By unzipping atoms in some residues, the protein-specific ESM model becomes a model at both residue and atom scales.

**Weaknesses:**

1. To me, the presentation, especially the experiments part, is not clear. Fro example, the authors use ‘for more details of …., readers can find them in …’ many times, but this really restrict me to understand the implementation details. A better way could be ‘following …, we fine tune … using ….’ In addition, please list the data size of downstream tasks.

2. About the ms-ESM model: during pre-training, what percentage of residues are unzipped?

3. Ablation: The pair-wise distance recovery is only used at atom scale and requires atom coordinates as inputs. How about removing this loss? What is the performance? Is this term necessary?

4. The performance on protein-molecule tasks: I can’t intuitively understand why ms-ESM can outperform two separate pre-trained models (one for protein, and one for molecule)? Basically, I think the capacity of two models should be greater than a single model. Is the comparison fair? Please provide more explanation.

**Questions:**

See weaknesses.

---

> ### Author Response · Authors · 2023-11-18
> **Response to Reviewer ZacV (part 1)**
>
> We thank reviewer ZacV for the helpful suggestions. In the following, we will address all your concerns regarding the paper presentation, implementation details, ablations, and intuitions. We have updated them in the revised manuscript and hope the replies can make our paper more clear. Any further comments are welcome!
>
> **Q1**: As many implementation details are omitted, the presentation in the experiments section is not clear enough.
>
> **A1**: Thanks for your kind notes. Previously, we cited corresponding papers for aligned details but did not include these details in our paper. **We have added these details in the revised manuscript to make it self-contained** and list them as follows:
> - Extra implementation details of protein-molecule tasks
>   - *Fine-tuning dataset.* Following ProSmith [1] (section 3.1), we finetune ms-ESM and all baseline models on dataset KM [2], Davis [3], and ESP [4] for enzyme-substrate affinity regression, drug-target affinity regression, and enzyme-substrate pair classification respectively. The KM dataset contains experimental affinity constants of 11,676 enzyme-substrate pairs. The Davis dataset provides 30,056 binding affinities for pairs of 72 drugs and 442 proteins. The ESP dataset consists of 68,754 positive/negative enzyme-substrate pairs with experimental evidence. We use the standard data split provided by ProSmith in fine-tuning.
>   - *Fine-tuning framework.* As mentioned in section 3.1, we use ProSmith's framework for a fair comparison. Specifically, the framework contains three main modules, i.e., molecule encoder, protein encoder, and fusion block. Two encoders extract features from proteins and molecules severally. The fusion block is a Transformer model, which is responsible for fusing protein and molecule features. The fused features are further used to regress the affinity values or predict binary affinity. We apply our model to ProSmith's framework by replacing both protein and molecule encoder with ms-ESM. We also provide the results of an XGBoost [12] variant of ProSmith, which removes the fusion block and uses simple concatenation for feature fusing. Note that we freeze both encoders in the experiments as suggested by ProSmith.
>   - *Fine-tuning hyperparameters.* We directly use the hyperparameters provided by ProSmith. Specifically, the fusion block for three tasks has 6 layers of Transformer whose hidden size is 768. The epoch number is 100 and the learning rate is 1e-5. The batch sizes of the three tasks are 12, 12, and 24. We use Adam [14] as the optimizer for ProSmith and GBDT [13] with 500 iterations as the predictors for XGBoost.
> - Extra implementation details for protein-only tasks
>   - *Fine-tuning dataset.* Following TAPE's protocol [5], we evaluate ms-ESM on secondary structure prediction and contact prediction tasks (section 3.2). Specifically, for secondary structure prediction, we use data from [6] as training and validation sets and use CB513 [7], CASP12 [8], and TS115 [9] as test sets. The training and validation sets are filtered at the 25% sequence identity threshold with these test tests. The final training, validation and three test sets have 8678, 2170, 513, 21, 115 protein sequences, respectively. For contact prediction tasks, we use training, validation, and test sets from ProteinNet [10] with training and validation sets filtered at the 30% sequence identity threshold. The final training, validation, and test sets have 20, 24, 13945 protein sequences.
>   - *Fine-tuning framework.* As suggested by TAPE, for both protein-only tasks, we use ms-ESM as the protein encoder. When doing secondary structure prediction, we use a linear output layer to predict the secondary structure each residue belongs to. When handling the contact prediction task, we use the attention from the last layer as features and then use a linear layer to predict whether these two residues have contact or not. Notably, both input of these two tasks is only protein sequences without structural information. Therefore, when using ms-ESM to handle these two tasks, we turn off the unzip.
>   - *Fine-tuning hyperparameters.* We set up all the hyperparameters aligned to TAPE. For secondary structure prediction, the epoch is 5,000, batch size is 10, and learning rate is 0.001. For contact prediction, the epoch is 5, batch size 64, and learning rate is 3e-5. We use AdamW [15] as the optimizer in secondary structure prediction and Adam in contact prediction.
>
> (To be continued.)

---

> ### Author Response · Authors · 2023-11-18
> **Response to Reviewer ZacV (part 2)**
>
> (To continue)
>
> - Extra implementation details for molecule-only tasks
>   - *Fine-tuning dataset.* We use the fine-tuning data of Uni-Mol [11] to evaluate the molecule understanding ability of ms-ESM (section 3.3). Specifically, we use QM8, QM9 datasets for molecular property regression and HIV, MUV datasets for molecular property classification, which have 21786, 133885, 41127, 93087 molecules, respectively. The data split is also provided by Uni-Mol.
>   - *Fine-tuning framework.* Following Uni-Mol, a special token, i.e., [CLS], also exists in ms-ESM. Similar to NLP/CV, we simply use the representation of [CLS] to represent the whole molecule, and then use a linear head for fine-tuning on downstream tasks. For each molecule, we use the 3D conformation provided by [11] as the input of ms-ESM. In the fine-tuning stage, we do not add noises to atom coordinates.
>   - *Fine-tuning hyperparameters.* For a fair comparison, we did not search the best hyperparameters. Instead, we set up all the hyperparameters aligned to Uni-Mol. Specifically, the batch sizes for these four tasks are 32, 128, 256, and 128. The learning rates are 1e-4, 1e-4, 5e-5, and 2e-5. The training epochs are 40, 40, 5, and 40. We use Adam optimizer for all the tasks.
>
> **Q2**: The data size of downstream tasks.
>
> **A2**: Thanks for pointing it out. We report it in the following table and **have added it to our revised manuscript** to make it clear. ESAR: Enzyme-Substrate Affinity Regression, DTAR: Drug-Target Affinity Regression, ESPC: Enzyme-Substrate Pair Classification, SSP: Secondary Structure Prediction, CP: Contact Prediction, MPR: Molecular Property Regression, MPC: Molecular Property Classification.
> |  | Protein-Molecule tasks |  |  | Protein-only tasks |  | Molecule-only tasks |  |  |  |
> | --- | --- | --- | --- | --- | --- | --- | --- | --- | --- |
> | task | ESAR | DTAR | ESPC | SSP | CP | MPR |  | MPC |  |
> | dataset | KM | Davis | ESP | NetSurfP‐2.0, CB513, CASP12, TS115 | ProteinNet  | QM8 | QM9 | HIV | MUV |
> | train | 8407 | 24045 | 49876 | 8678 | 20 | 17428 | 107108 | 32901 | 74469 |
> | val | 934 | 3006 | 5540 | 2170 | 24 | 2179 | 13388 | 4113 | 9309 |
> | test | 2335 | 3005 | 13336 | 513/21/115 | 13945 | 2179 | 13389 | 4113 | 9309 |
> | total | 11676 | 30056 | 68754 | 11497 | 13989 | 21786 | 133885 | 41127 | 93087 |
>
>
> **Q3**: During pre-training, what percentage of residues are unzipped?
>
> **A3**: Thanks for pointing it out. Due to the page limitation, we report it in Table 8 of Appendix A: 6.5% of residues are unzipped as the main experimental setting. We have added it with other details in the main text.
>
> **Q4**: The pair-wise distance recovery is only used at atom scale and requires atom coordinates as inputs. How about removing this loss? What is the performance? Is this term necessary?
>
> **A4**: Thank you very much for your good questions. Actually, the pair-wise distance recovery loss is the most widely used pretraining objective for captureing geometric molecular information[11][19]. As you suggested, we did a quick run by removing it for ablation and the results are shown in Table A and we also have added them to our revised manuscript.
>
> - Results: As shown in Table A, when we do not apply pair-wise distance recovery loss, the performance of ms-ESM shows a significant decrease in the protein-molecule task.
> - Analysis:
>   - For molecular representation learning
>     - Without using pair-wise distance recovery loss, the training loss of ms-ESM on the atomic scale is only masked atom type prediction loss. But more than half of the atoms in a molecule are carbon atoms, which makes the model learn very little information on the atomic scale using only masked atom type prediction.
>   - For modeling unzipped residues
>     - Without using pair-wise distance recovery loss, ms-ESM will not be able to learn structural information from the unzipped residues. ms-ESM can only learn the mapping from residues to atoms' types. Since there are only a few types of residues, and the mapping is also fixed for a given residue, this training task is too easy to make the model learn meaningful information.
>
> Table A: Ablation study on pair-wise distance recovery loss
> |  | MSE ↓ | R^2 ↑ |
> | --- | --- | --- |
> | Vanilla ms-ESM | 0.627 | 0.546 |
> | w/o pair-wise distance recovery loss | 0.645 (+0.018) | 0.533(-0.013) |

---

> ### Author Response · Authors · 2023-11-18
> **Response to Reviewer ZacV (part 3)**
>
> **Q5**: Can’t intuitively understand why ms-ESM can outperform two separate pre-trained models (one for protein, and one for molecule)? Basically, I think the capacity of two models should be greater than a single model.
>
> **A5**:
> - Yes, the capacity of two separate models is larger than one unified model. ms-ESM performs better because **the benefits of unified modeling inside one model (with specific designs) make it learn the unified context jointly and implicitly, which outweighs its shortcomings in capacity.**
> - The above observation has also been found in multi-lingual learning and multi-modal learning areas. Specifically, [18] shows that a unified multi-lingual machine translation model (one model for all language pairs) can outperform translation models trained separately. [17] demonstrates that modeling vision and language directly inside one model can beat methods that fuse the two modalities later.
> - Moreover, visualization in our paper (Figure 4) also indicates that our unified ms-ESM can provide more harmonious protein/molecule features than two separate models, i.e., EMS-2 and Uni-Mol.
>
> **Q6**: Is the comparison in protein-molecule tasks fair?
>
> **A6**:
> - Yes, the comparison is fair. As mentioned in Section 3.1, we follow the benchmark protocol from ProSmith [1] to evaluate ms-ESM and all the baseline methods. In other words, the settings of all the models being compared are well-aligned.
> - Smaller ms-ESM outperforms two larger models, which proves the benefit of unified modeling even more significantly. Specifically, ms-ESM is at a disadvantage of the model capacity compared to ESM-2 + Uni-Mol (35M < 35M + 48M). However, Table 1, 2, and 3 in our paper clearly show that ms-ESM beats ESM-2 + Uni-Mol.
>
> [1] Kroll A, Ranjan S, Lercher M J. A multimodal Transformer Network for protein-small molecule interactions enhances drug-target affinity and enzyme-substrate predictions. bioRxiv 2023
>
> [2] Kroll A, Engqvist M K M, Heckmann D, et al. Deep learning allows genome-scale prediction of Michaelis constants from structural features. PLoS biology 2021
>
> [3] Davis M I, Hunt J P, Herrgard S, et al. Comprehensive analysis of kinase inhibitor selectivity. Nature biotechnology 2011
>
> [4] Kroll A, Ranjan S, Engqvist M K M, et al. A general model to predict small molecule substrates of enzymes based on machine and deep learning. Nature Communications 2023
>
> [5] Rao R, Bhattacharya N, Thomas N, et al. Evaluating protein transfer learning with TAPE. NeurIPS 2019
>
> [6] Klausen M S, Jespersen M C, Nielsen H, et al. NetSurfP‐2.0: Improved prediction of protein structural features by integrated deep learning. Proteins: Structure, Function, and Bioinformatics 2019
>
> [7] Cuff J A, Barton G J. Evaluation and improvement of multiple sequence methods for protein secondary structure prediction. Proteins: Structure, Function, and Bioinformatics 1999
>
> [8] Moult J, Fidelis K, Kryshtafovych A, et al. Critical assessment of methods of protein structure prediction (CASP)—Round XII. Proteins: Structure, Function, and Bioinformatics 2018
>
> [9] Yang Y, Gao J, Wang J, et al. Sixty-five years of the long march in protein secondary structure prediction: the final stretch?. Briefings in bioinformatics 2018
>
> [10] AlQuraishi M. ProteinNet: a standardized data set for machine learning of protein structure. BMC bioinformatics 2019
>
> [11] Zhou G, Gao Z, Ding Q, et al. Uni-Mol: a universal 3D molecular representation learning framework. ICLR 2023
>
> [12] Chen T, Guestrin C. Xgboost: A scalable tree boosting system. KDD 2016
>
> [13] Ke G, Meng Q, Finley T, et al. Lightgbm: A highly efficient gradient boosting decision tree. NeurIPS 2017
>
> [14] Kingma D P, Ba J. Adam: A method for stochastic optimization. arXiv 2014
>
> [15] Loshchilov I, Hutter F. Decoupled weight decay regularization. arXiv 2017
>
> [16] Dong L, Yang N, Wang W, et al. Unified language model pre-training for natural language understanding and generation. NeurIPS 2019
>
> [17] Jang J, Kong C, Jeon D, et al. Unifying vision-language representation space with single-tower transformer. AAAI 2023
>
> [18] Johnson M, Schuster M, Le Q V, et al. Google’s multilingual neural machine translation system: Enabling zero-shot translation. TACL 2017
>
> [19] Yu Q, Zhang Y, Ni Y, et al. Unified Molecular Modeling via Modality Blending. arXiv 2023

---

> ### Author Response · Authors · 2023-11-21
> **Response to Reviewer ZacV (part 4)**
>
> Dear Reviewer ZacV, we are grateful for your insightful feedback, which has been instrumental in improving our manuscript.
>
> Please let us know if we have adequately addressed your concerns. We are always open to further suggestions.
>
> Thank you for your time and consideration.

---

> ### Author Response · Authors · 2023-11-22
> **Seeking Feedback on Our New Response**
>
> Thank you for your thoughtful review. We have addressed the concerns you raised in our response. Could you kindly confirm if our explanations adequately address your points? We're open to further discussion and ready to address any remaining questions you may have. Feel free to reach out for a quick discussion.

---

> > ### Comment · Reviewer_ZacV · 2023-11-22
> >
> > Thanks a lot for the authors' detailed response. Most of my concerns are addressed. I also read other reviewers' comments and agree that this paper can be further improved.
> >
> > Overall, I decided to keep my score at 5 and hope the author can improve the presentation, add all of these clarification details, and emphasize the contribution, etc in the main text in your next version.

---

> > > ### Author Response · Authors · 2023-11-23
> > >
> > > Thank you for reading our response and providing feedback. We will also make improvements to address some issues in the final version of the paper to ensure a more rigorous and clear presentation. If you have any further questions, please feel free to continue the conversation.

---

### Author Response · Authors · 2023-11-18
**Response to all the reviewers and area chairs (part 1)**

Thanks for all your efforts in reviewing this paper. We want to express our sincere appreciation to the reviewers and the area chairs for their insightful, and constructive comments. We will first address the common concerns in the general response and then respond to specific concerns raised by different reviewers respectively. We have updated our submission (with red text) to clarify our approach. For your convenience, we also list the changes here:
- In Appendix D, we supplement some experimental results on more molecule-only tasks, as suggested by Reviewer VsVJ.
- In Appendix E, we supplement more results on ablation study, as suggested by Reviewer ZacV and 54HD.
- In Appendix B, we add more fine-tuning details to make our paper more self-contained, as suggested by Reviewer ZacV.
- In Appendix C, we discuss the connection between our work and previous molecular modeling methods, as suggested by Reviewer 54HD.
- In Appendix A, we elaborate more details of model design, including ORDER procedure (Reviewer p5PD), the size of fine-tuning datasets, and unzip ratio (Reviewer ZacV).
- In Section 3.2, we provide explanations on the data leakage issue raised by the Reviewer 54HD.

Briefly, reviewers raised questions mainly from the following perspectives:
- Model performance (Reviewer 54HD, VsVJ, ZacV, p5PD)
- Technical novelty (Reviewer 54HD, p5PD)
- Model design (Reviewer 54HD, p5PD, ZacV)
- Ablation study (Reviewer 54HD, ZacV)
- Paper writing (Reviewer 54HD, ZacV)
- Data leakage (Reviewer 54HD)

---

### Author Response · Authors · 2023-11-18
**Response to all the reviewers and area chairs (part 2)**

Here we will respond to two common concerns regarding model performance and technical novelty as follows:

**Q1**: The unified pre-training appears to diminish performance on tasks focused solely on proteins (Table 4, Table 5 in our paper) or molecules  (Table 6 in our paper).

**A1**: Thanks for the question. However, it appears that there might be some misunderstandings regarding the proposed ms-ESM.
- We want to reclarify our motivation: **the unified modeling of multi-scale information in ms-ESM is to obtain improvements on protein-molecule tasks instead of all tasks, including solely on proteins and molecules.** We give the results **solely on proteins and molecules to verify whether the unified ms-ESM lost too much information on the two kinds of separated tasks.**
- Results show that ms-ESM 1) outperforms ESM-2+Uni-Mol on the protein-molecule task with a large margin, 2) obtains comparable accuracies to ESM on protein-only tasks (exhibits both highs and lows rather than a uniform decline) and 3) performs reasonably close  to Uni-Mol on the molecule-only task. We give more details and discussions as follows:
  - **ms-ESM retains a completely comparable ability to ESM-2 in protein-only tasks.** Note that for fine-tuning, ms-ESM uses the same setting as ESM-2 (the same hyper-parameters, the same inputs, and the same training process).
    - On the 9 metrics of contact prediction (Table 4 in our paper), the average accuracy of ms-ESM is slightly higher than ESM-2 by 0.04% (35% vs 34.6%). i) In short-range contact prediction scenarios, the ms-ESM model's performance can be slightly higher than ESM-2. ii) In long-range contact prediction scenarios, the ms-ESM model's performance is slightly lower than ESM-2.
    - On the 6 metrics of secondary structure prediction (Table 5 in our paper), the average level of ms-ESM is slightly lower than ESM-2 by 1% (71% vs 72%).
  - **ms-ESM performs reasonably closely with Uni-Mol w/o H in molecule-only tasks.** Note that for both pretraining and finetuning, we adopt the same setting as Uni-Mol w/o H (hydrogen atoms were completely removed during the pre-training process). We also test the model performance on more molecular tasks (shown in Table A). **Uni-Mol [1] is a strong baseline specifically designed for geometric molecular modeling.** The performance gap between ms-ESM and Unimol is reasonable because:
    - To align with the parameter scale of ESM-2, ms-ESM uses a smaller and shallower model than Uni-Mol (35M vs 47M, 15 layers vs 12 layers).
    - Uni-Mol is  a model specifically designed for geometric molecular modeling, which adopts more pre-training objectives for capturing geometric molecular information. ms-ESM only employs the most widely used one and does not include objectives like 3D Position Recovery in unified multi-scale modeling for the purpose of efficient pre-training.

  To further show that the construction of **the unified model does not compromise the model's molecular representation learning capability**, we train a molecule-only ms-ESM to determine what is the cause of performance degradation. The pre-training of the molecule-only ms-ESM uses exactly the same settings as ms-ESM, except that only molecular data is used for the pre-training data.

  The performance of the molecule-only model is shown in Table A. Results show that the molecule-only model's performance on molecular tasks is still inferior to Uni-Mol, which suggests that the difference in model size and the lack of training objectives may indeed be responsible for the performance disadvantage of ms-ESM. 2) The molecule-only model can't outperform ms-ESM, which suggests that protein-molecule joint training is **NOT** the cause of performance degradation.

- Although ms-ESM cannot outperform ESM-2 and Uni-Mol in protein- and molecule-only tasks, **it obtains consistent and significant improvements to ESM-2+Uni-Mol in all protein-molecule tasks** (using the same fine-tuning and evaluating protocol, results are shown in Table 1, Table 2, Table 3 in our paper), which verifies the motivation of this paper that ms-ESM unified multi-scale modeling benefits protein-molecule tasks.

Table A: New experimental results on more molecule-only tasks
|  | QM7 ↓ | QM8 ↓ | QM9 ↓ | BACE↑ | BBBP↑ | TOX21↑ | PCBA↑ | SIDER↑ | HIV↑ | MUV↑ |
| --- | --- | --- | --- | --- | --- | --- | --- | --- | --- | --- |
| Uni-Mol w/o H | 58.8678 | **0.0160** | **0.00540** | 83.2 | **71.52** | **78.92** | **88.12** | 57.71 | **78.30** | 72.0 |
| ms-ESM | 60.8755 | 0.0166 | 0.00590 | **83.52** | 67.41 | 75.39 | 86.15 | **63.59** | 74.90 | **72.6** |
| molecule-only ESM | **56.5399** | 0.0170 | 0.00600 | 82.89  | 65.19  | 75.23  | 87.17  | 63.32  | 76.12  | 72.4  |

---

### Author Response · Authors · 2023-11-18
**Response to all the reviewers and area chairs (part 3)**

**Q2**: Based on existing techniques like Transformer architecture and RoPE, ms-ESM's technical novelty is limited.

**A2**: Yes, ms-ESM is based on several existing techniques. But the novelty of ms-ESM lies in three-folds:
- **Constructing a model, i.e., ms-ESM, which can process proteins and molecules universally, is the main contribution of our work.** Although code-switching has been widely explored in the NLP domain, we occasionally realize that it is quit suitable for efficiently unified molecular modeling, after our initial and aborted attempts that jointly model proteins and small molecules in a full atom way.
- **Unified multi-scale molecular modeling is the cutting edge of AI biological application.** Specifically, after the submission DDL of ICLR 2024, two impressive papers, i.e., **RoseTTAFold All-Atom [2] and AlphaFold-latest [3], adopt similar modeling strategies showing great performance** in wide application tasks, including molecule docking, covalent modified structure prediction, and binder design.
- **Adopting existing techniques to our multi-scale unified molecular modeling is non-trivial.** Specifically, using code-switching in molecular modeling is not a straightforward idea. We find its benefits only after our failed inefficient full atom modeling. Moreover, merging two position encodings to accurately describe the complex multi-scale relationships also requires careful designing, e.g., sharing RoPE with atoms from the same residue to avoid introducing ill-defined residue scale position relationships inside the residue.

[1] Lample G, Conneau A. Cross-lingual language model pretraining. arXiv 2019.

[2] Krishna R, Wang J, Ahern W, et al. Generalized Biomolecular Modeling and Design with RoseTTAFold All-Atom. bioRxiv 2023.

[3] Google DeepMind AlphaFold Team and Isomorphic Labs Team. Performance and structural coverage of the latest, in-development AlphaFold model. Google DeepMind. https://deepmind.google/discover/blog/a-glimpse-of-the-next-generation-of-alphafold/ : 2023-10.

---

### Meta-Review · Area_Chair_ySVQ · 2023-12-11

**Metareview:**

The paper introduces msESM, a multi-scale unified molecular model designed to enhance the applications of proteins. The model, which uses pre-training on multi-scale code-switch protein sequences and multi-scale position encoding, outperforms previous methods in protein-molecule tasks and matches the performance of leading models in protein-only and molecule-only tasks.

Despite significant improvements made by the authors during the discussion period, reviewers remained steadfast in their negative evaluations. Their concerns included a lack of technical innovation, unclear experimental contribution, and missing details and explanations.

**Justification For Why Not Higher Score:**

Despite significant improvements made by the authors during the discussion period, reviewers remained steadfast in their negative evaluations. Their concerns included a lack of technical innovation, unclear experimental contribution, and missing details and explanations.

**Justification For Why Not Lower Score:**

N/A

---

### Decision · Program_Chairs · 2024-01-16

Reject